# Structural analysis identifies an escape route from the adverse lipogenic effects of liver X receptor ligands

Anna Y. Belorusova[1]*, Emma Evertsson[1]*, Daniel Hovdal[2], Jenny Sandmark[3], Emma Bratt [4], Ingela Maxvall[5], Ira G. Schulman [6], Peter Åkerblad[7,9] & Eva-Lotte Lindstedt[8]

Liver X receptors (LXRs) are attractive drug targets for cardiovascular disease treatment due to their role in regulating cholesterol homeostasis and immunity. The anti-atherogenic properties of LXRs have prompted development of synthetic ligands, but these cause major adverse effects—such as increased lipogenesis—which are challenging to dissect from their beneficial activities. Here we show that LXR compounds displaying diverse functional responses in animal models induce distinct receptor conformations. Combination of hydrogen/deuterium exchange mass spectrometry and multivariate analysis allowed identification of LXR regions differentially correlating with anti-atherogenic and lipogenic activities of ligands. We show that lipogenic compounds stabilize active states of LXRα and LXRβ while the anti-atherogenic expression of the cholesterol transporter ABCA1 is associated with the ligand-induced stabilization of LXRα helix 3. Our data indicates that avoiding ligand inter-action with the activation helix 12 while engaging helix 3 may provide directions for devel-opment of ligands with improved therapeutic profiles.

[1] Medicinal Chemistry, Respiratory, Inflammation and Autoimmunity, BioPharmaceuticals R&D, AstraZeneca, Gothenburg, Sweden. [2] Preclinical and Translational PK & PKPD, Cardiovascular, Renal and Metabolism, BioPharmaceuticals R&D, AstraZeneca, Gothenburg, Sweden. [3] Structure, Biophysics & Fragment Based Lead Generation, Discovery Sciences, R&D, AstraZeneca, Gothenburg, Sweden. [4] Medicinal Chemistry, Cardiovascular, Renal and Metabolism, BioPharmaceuticals R&D, AstraZeneca, Gothenburg, Sweden. [5] Translational Science and Experimental Medicine, Cardiovascular, Renal and Metabolism, BioPharmaceuticals R&D AstraZeneca, Gothenburg, Sweden. [6] Department of Pharmacology, University of Virginia, Charlottesville, VA, USA. [7] Bioscience Heart Failure, Cardiovascular, Renal and Metabolism, Biopharmaceuticals R&D AstraZeneca, Gothenburg, Sweden. [8] Early Cardiovascular, Renal and Metabolism, BioPharmaceuticals R&D, AstraZeneca, Gothenburg, Sweden. [9]Present address: Albireo Pharma, Gothenburg, Sweden. *email: anna.belorusova@astrazeneca.com; emma.evertsson@astrazeneca.com

The liver X receptors (LXRs) alpha (LXRα, NR1H3) and beta (LXRβ, NR1H2) are members of the nuclear hormone receptor (NHR) superfamily of transcription factors that function as heterodimers with the retinoid X receptor (RXR) and activate or repress gene expression in response to binding of cholesterol derivatives or synthetic ligands[1–3]. Historically LXRs are considered attractive drug targets for treatment of atherosclerotic cardiovascular disease because they regulate transcriptional networks that maintain cholesterol homeostasis[4–6]. The LXRs induce macrophage, liver, and intestine levels of the ATP-binding cassette cholesterol transporters ABCA1, ABCG1[7–9], ABCG5, and ABCG8[10–12], as well as apolipoprotein E[13], which results in enhanced cholesterol efflux from macrophages and excretion of cholesterol from the liver and intestine, leading to a net movement of cholesterol out of the body[14–18]. Treatment with synthetic LXR ligands leads to regression and stabilization of atherosclerotic lesions in animal models of atherosclerosis[19–21]. The ability of LXR ligands to modulate immune and inflammatory responses in macrophages likely contributes to their anti-atherogenic potential[22].

Despite decades of intensive efforts, development of anti-atherogenic LXR compounds has been hindered by their lipogenic effects[23,24] as LXRs directly regulate expression of several lipogenic genes including those encoding sterol regulatory element-binding protein 1c, fatty acid synthase[25], and stearoyl-coenzyme A desaturase 1[26]. Treatment with pan-LXR agonists results in elevated hepatic fatty acid and triglyceride (TG) synthesis, leading to hypertriglyceridemia and hepatic steatosis in rodent models[23,27]. Since the adverse lipogenic effects are considered to be largely driven by LXRα[23,28], drug discovery efforts have focused on developing LXRβ-selective compounds. Indeed, pharmacological activation of LXRβ alone was shown to ameliorate the cholesterol overload phenotype in mouse models of atherosclerosis[29,30]. Reaching isoform selectivity by rational structure-based drug design has proven to be difficult since the ligand-binding pockets (LBPs) of LXRα and LXRβ are virtually identical[31–33]. Nevertheless, some degree of selectivity in binding and functional assays was achieved for a number of compounds[33–36], including two ligands displaying reduced lipogenic effects and reaching clinical evaluation: WAY-252623 (LXR-623)[37–39] and BMS-852927[40,41]. However, even optimized compounds retain substantial affinity for LXRα, and therefore it is questionable whether their reduced hepatic activities could be explained solely by modestly improved LXRβ selectivity.

It could be hypothesized that LXR compounds with improved therapeutic profiles alter receptor conformation in a different way than ligands inducing strong adverse effects. Transcriptional responses of LXRs are known to directly result from ligand-induced conformational changes in the receptor: similarly to other NHRs, in the absence of ligands or in the presence of antagonists, LXRs form complexes with corepressors, while binding of agonist ligands forces the C-terminal helix H12 of the receptor to fold into the active state that creates an activation function-2 (AF-2) coactivator-binding surface[42]. Altered conformation and structural dynamics of the receptor can therefore be directly related to the nature of ligand and result in selective recruitment of transcriptional coregulators to target gene promoters.

To explore how pharmacologically different classes of LXR compounds affect LXRα and LXRβ structural dynamics, we measured conformational changes of both receptors by hydrogen/deuterium exchange mass spectrometry (HDX-MS). HDX-MS is a pseudostructural method that reveals perturbations in protein dynamics upon ligand binding at the peptide level, and therefore can be used to evaluate interaction details. HDX-MS has emerged as an important tool for the development of therapeutics[43–45] and

has a high potential for predicting functional activity of ligands when combined with statistical tools and functional data[46–48]. In this study, we integrated HDX-MS analysis and phenotypic screening in mice by using a multivariate analysis strategy to identify differential LXR regions associated with ligand-induced elevation of TG and Abca1 mRNA levels. Our results demonstrate that conformations of both LXRα and LXRβ are differentially perturbed by compounds displaying high activities over low-activity ligands. Different protein regions were found to be associated with TG and Abca1 induction by LXR ligands, indicating that HDX-MS can be used for identification of vectors for the rational design of compounds with improved therapeutic profiles.

## Results

**Assembling a collection of LXR ligands.** To identify conformational patterns specifically related to positive and adverse effects of LXR ligands, we selected a small set of LXR agonists displaying a broad range of profiles in an unbiased C57BL/6 in vivo mouse screen from an anti-atherogenic LXR-ligand optimization program. We have previously reported that the LXR agonist AZ876 (Fig. 1) inhibits the progression of atherosclerosis in mice without inducing liver steatosis or hypertriglyceridemia when given at low doses[49,50]. In this study, we compared AZ876 with structurally related compounds AZ1–6 (Supplementary Fig. 1a). These compounds have a central cyclic sulfone amide scaffold with lipophilic substituents on both sides identical to AZ876, except for the tert-butyl substituent replaced by a propyl group in AZ1–3 and AZ6. The third substituent from the core is a larger elongated amine linker that is different in AZ1–6. The amine linker is mostly lipophilic with a polar group at the far end. We have also included compounds AZ7–9 representing a structurally related series where the central scaffold ring is replaced by a maleimide core (Supplementary Fig. 1b). The AZ compounds were compared with previously described LXR ligands including the full pan-agonists T0901317[23] and WAY-254011[51], the LXRα-selective agonist F1 developed by Laboratoires Fournier S.A.[52], and agonists with improved therapeutic windows GW3965[53,54], WAY-252623 (LXR-623)[37–39], and BMS-852927[40,41] (Fig. 1). We have also included a low-affinity endogenous ligand 24(S),25-epoxycholesterol (24,25EC)[3] into in vitro and biophysical assays.

**In vitro characterization of LXR ligands.** All compounds were profiled for binding affinities to LXRα and LXRβ ligand-binding domains (LBDs) and for LXRα/LXRβ agonist activities in a mammalian Gal4 one hybrid reporter gene assay (Table 1). All compounds demonstrated nanomolar affinities to the LXRβ LBD. The compounds AZ1, AZ3, and BMS-852927 have the highest affinities of 2, 4, and 3 nM, and the compounds AZ6, AZ7, and 24,25EC have the lowest affinities of 161, 105, and 133 nM, respectively. The binding affinities to LXRα varied more widely from low nanomolar (6 nM for AZ876, 10 nM for F1) to low micromolar (2.1, 1.6, and 1 μM for compounds AZ5, AZ6, and AZ7, respectively). Among the tested compounds, ligands AZ1–5 and AZ7 displayed more than tenfold LXRβ binding selectivity.

All compounds except 24,25EC and BMS-852927 displayed agonist activities in the Gal4 reporter gene assays. No signal was observed for the latter two compounds although they were previously characterized as partial agonists in similar in vitro assays[40,55]. Together with T0901317, compounds AZ876 and AZ3 displayed a potency below 100 nM in both LXRα- and LXRβ-based assays (Table 1). Potency-based LXRβ selectivity was observed for AZ2, AZ3, and AZ5 (11-, 9-, and 11-fold selectivity over LXRα, respectively), while AZ876 and F1 were LXRα selective (12- and 7-fold selectivity over LXRβ, respectively).

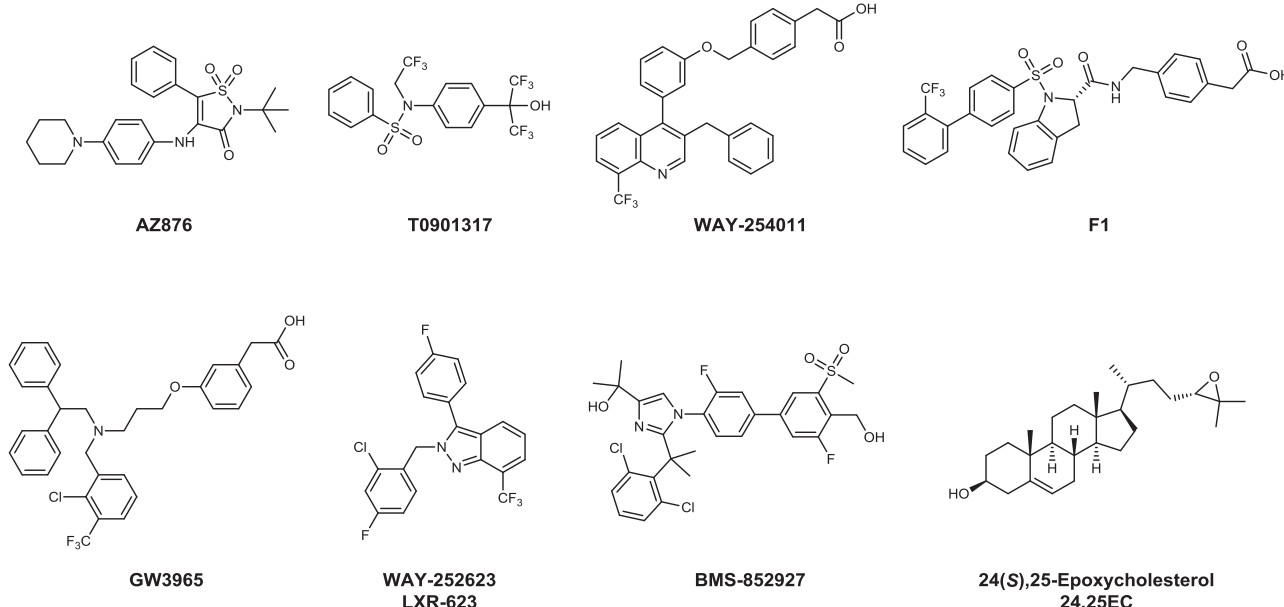

**Fig. 1** Examples of LXR compounds used in this study.

**Table 1 In vitro properties of ligands.**

| | Receptor-binding affinity | | LXRα transactivation | | LXRβ transactivation | |
|---|---|---|---|---|---|---|
| | pKi LXRα | pKi LXRβ | pEC$_{50}$ | Agonist efficacy, % | pEC$_{50}$ | Agonist efficacy, % |
| AZ1 | 7.04 | 8.6 | 6.96 | 102 | 7.51 (0, $n = 2$) | 76.0 (0, $n = 2$) |
| AZ2 | 7.04 (0.16, $n = 7$) | 8.43 (0.07, $n = 7$) | 7.11 (0.12, $n = 7$) | 111.4 (10.7, $n = 7$) | 8.06 (0.13, $n = 7$) | 97.3 (12.2, $n = 7$) |
| AZ3 | 6.3 (0.08, $n = 3$) | 7.76 (0.15, $n = 3$) | 6.09 (0.37, $n = 4$) | 142.5 (37.7, $n = 4$) | 7.12 (0.34, $n = 4$) | 94.3 (18.0, $n = 4$) |
| AZ4 | 6.45 (0.48, $n = 2$) | 7.48 (0.36, $n = 2$) | 6.37 (0.03, $n = 2$) | 103.0 (9.9, $n = 2$) | 6.83 (0.13, $n = 4$) | 110.0 (0, $n = 4$) |
| AZ5 | 5.67 (0.28, $n = 2$) | 7.11 (0.19, $n = 2$) | 5.97 (0.23, $n = 2$) | 60.5 (21.9, $n = 2$) | 7.01 (0.02, $n = 2$) | 66.5 (10.6, $n = 2$) |
| AZ6 | 5.8 (0.25, $n = 5$) | 6.79 (0.29, $n = 5$) | 5.91 (0.23, $n = 6$) | 87.8 (5.0, $n = 6$) | 6.21 (0.15, $n = 7$) | 95.3 (2.9, $n = 7$) |
| AZ7 | 5.97 (0.14, $n = 10$) | 6.98 (0.19, $n = 10$) | 5.96 (0.12, $n = 10$) | 85.3 (8.7, $n = 10$) | 6.27 (0.05, $n = 11$) | 104.5 (13.5, $n = 11$) |
| AZ8 | 6.71 | 7.32 | 5.76 (0.08, $n = 2$) | 91.5 (10.6, $n = 2$) | 5.89 (0.07, $n = 3$) | 102.0 (6.9, $n = 3$) |
| AZ9 | 6.24 | 7.09 | 6.33 (0.14, $n = 3$) | 90.3 (9.0, $n = 3$) | 6.55 (0.11, $n = 4$) | 99.0 (1.4, $n = 4$) |
| AZ876 | 8.22 (0.25, $n = 3$) | 7.98 (0.09, $n = 3$) | 8.25 (0.21, $n = 4$) | 101.0 (17.1, $n = 3$) | 7.15 (0.11, $n = 5$) | 107.4 (14.1, $n = 5$) |
| T0-091317 | 7.15 | 7.58 | 7.4 (0.12, $n = 71$) | 100.8 (7.4, $n = 4$) | 7.19 (0.13, $n = 67$) | 95.5 (6.4, $n = 2$) |
| WAY-254011 | 7.34 | 7.95 | 6.86 | 85 | 7.23 | 69.0 (0, $n = 2$) |
| F1 | 7.99 | 7.44 | 6.25 | 84 | 5.41 | 105 |
| GW3965 | 6.8 (0.24, $n = 5$) | 7.62 (0.29, $n = 5$) | 5.96 (0.18, $n = 15$) | 93.5 (15.7, $n = 12$) | 6.46 (0.18, $n = 22$) | 77.1 (6.2, $n = 19$) |
| LXR-623 | 6.31 (0.19, $n = 4$) | 7.02 (0.08, $n = 4$) | 5.59 (0.09, $n = 4$) | 102.3 (18.7, $n = 4$) | 5.73 (0.06, $n = 4$) | 94.3 (24.3, $n = 4$) |
| BMS-852927 | 7.69 (0.13, $n = 3$) | 8.58 (0.15, $n = 3$) | <4.7 | NA | <4.7 | NA |
| 24,25EC | 6.56 (0.2, $n = 2$) | 6.88 (0.15, $n = 2$) | <4.3 | NA | <4.3 | NA |

pEC$_{50}$ values describe the negative logarithm of the concentrations at which tested compounds reach their half-maximal signal. Agonist efficacy describes the maximal achieved activation compared with a reference full agonist compound. The numbers in parentheses are the standard deviations and number of independent experiments. Values without standard deviations were obtained in experiments run once. Values obtained in each individual experiment are derived from the average of three (binding affinity) or four (transactivation) measurements

Taken together, in vitro profiling demonstrates that our collection includes compounds with a wide range of LXR binding affinities, isoform selectivity, and in vitro potencies.

**Classification of LXR ligands based on their beneficial and adverse effects in vivo.** To build and compare multivariate models for positive and adverse pharmacological effects of LXR ligands, we ranked the compounds according to their displayed in vivo potencies. Beneficial anti-atherogenic effects of LXR ligands are thought to be associated with induced levels of the ABCA1 transporter in the intestine, liver, and macrophages within the atherosclerotic lesions. Higher signal-to-noise ratio of intestinal *Abca1* mRNA increase is typically observed upon LXR-ligand administration compared with the gene expression levels in blood, probably due to higher local concentration of ligands as opposed to their systemic exposure. We therefore chose to rank

LXR ligands based on intestinal *Abca1* induction. For the adverse effects, we focused on induced plasma TG levels that largely reflect the effect of ligands on hepatic gene expression.

In the C57BL/6 mice, all tested LXR ligands were found to increase intestinal *Abca1* transcript levels above basal (Supplementary Fig. 2). To compare potencies of compounds tested at different doses, the assessment of intestinal *Abca1* induction was performed by using a population approach where the compounds were assumed to have the same maximal effect ($E_{max}$). The median effective dose ($ED_{50}$) values were determined by simultaneously fitting all data with test compounds as a categorical covariate. AZ6 was selected as the reference drug, and all the other compounds were compared with it. In general, estimates with satisfactory precision were obtained (RSE < 30%) and no marked trends in the diagnostic plots were observed (Supplementary Fig. 3). AZ6 and compounds with higher potencies were assigned to the first class of ligands (high ABCA1

**Table 2 Pharmacodynamic parameters of drug-induced intestinal Abca1 transcript levels and plasma triglycerides.**

| Ligand | Abca1 mRNA induction in intestine | | | | TG induction in plasma | | | | |
|---|---|---|---|---|---|---|---|---|---|
| | $E_{max}$ (fold induction) | $ED_{50}$, µmol kg$^{-1}$ | Residual error[a] | Attributed class | $\gamma$ | $m$ | $C_2$, µM | Residual error[b] | Attributed class |
| All | 15.1 (6) | | 0.38 (7) | | 0.528 (23) | – | – | 0.410 (9) | |
| AZ1 | | <30[c] | | High ABCA1 | | 3.88 (30) | 0.214 (58) | | Lipogenic |
| AZ2 | | 9.64 (14) | | High ABCA1 | | 9.28 (63) | 0.041 (120) | | – |
| AZ3 | | 1.49 (14) | | High ABCA1 | | 0.897 (20) | 3.42 (38) | | – |
| AZ4 | | 5.65 (12) | | High ABCA1 | | 1.47 (28) | 1.35 (54) | | – |
| AZ5 | | 17.2 (19) | | Low ABCA1 | | n.d. | n.d. | | – |
| AZ6 | | 11.3 (12) | | High ABCA1 | | n.d. | n.d. | | Non-lipogenic |
| AZ7 | | <30[c] | | High ABCA1 | | n.d. | n.d. | | Non-lipogenic |
| AZ8 | | 341 (9) | | Low ABCA1 | | n.d. | n.d. | | Non-lipogenic |
| AZ9 | | 32.2 (11) | | Low ABCA1 | | n.d. | n.d. | | Non-lipogenic |
| AZ876 | | 0.956 (10) | | High ABCA1 | | 3.44 (49) | 0.269 (93) | | Lipogenic |
| T0901317 | | 4.11 (9) | | High ABCA1 | | 0.784 (17) | 4.41 (33) | | Lipogenic |
| WAY-254011 | | <30[c] | | High ABCA1 | | 1.94 (4) | 0.794 (8) | | Lipogenic |
| F1 | | <30[c] | | High ABCA1 | | 12.1 (38) | 0.025 (72) | | Lipogenic |
| GW3965 | | 0.969 (22) | | High ABCA1 | | 0.244 (18) | 40.2 (33) | | Non-lipogenic |
| LXR-623 | | 31.5 (13) | | Low ABCA1 | | n.d. | n.d. | | Non-lipogenic |
| BMS-852927 | | 2.10 (48) | | High ABCA1 | | 0.066 (18) | 477 (35) | | Non-lipogenic |

$E_{max}$: maximal fold of Abca1 induction, $ED_{50}$: dose generating 50% of maximal fold Abca1 induction, $\gamma$: amplification factor influencing the slope of the drug concentration versus TG increase plot, $m$: scaling factor shifting the amplitude of the drug concentration versus TG increase plot, $C_2$: drug concentration corresponding to a twofold increase in TG levels. The numbers in parentheses are the % relative standard errors of the parameter estimates
n.d. not determined
[a]Multiplicative residual error
[b]Additive residual error
[c]$ED_{50}$ could not be determined since no dose response was generated and the observed fold induction was in the range of the maximal Abca1 mRNA increase

inducers, $ED_{50} \leq 11.3$ µmol kg$^{-1}$), while compounds AZ5, AZ8, AZ9, and LXR-623 with weaker potencies were assigned to the second class (low ABCA1 inducers) (Table 2). Compounds WAY-254011, AZ1, AZ7, and F1 were included in the "high ABCA1" class of ligands. Their $ED_{50}$ values could not be determined as no sufficient dose–response data were available; however, they induced Abca1 mRNA levels in the range of the maximal observed increase as opposed to low ABCA1 inducers for which no $E_{max}$ was reached.

Plasma TG levels induced by studied LXR ligands were assessed as a function of plasma compound concentration and the concentration–response relationship was described by using a log-linear function. In general, less than a twofold increase in plasma TG was observed (Supplementary Fig. 4). Compound ranking was made based on the estimated drug concentration needed to generate a twofold increase from baseline ($C_2$), although the assessment was obstructed by the large relative baseline variability in TG levels (Supplementary Fig. 5), and therefore, poor precisions (RSE > 30%) in parameter estimates were generally obtained (Table 2). As no trends toward increased TG levels were observed for compounds AZ6–9 and LXR-623 within the explored concentration range, these ligands were assigned to a "non-lipogenic" class together with GW3965 ($C_2 = 40.2$ µM) and BMS-852927 ($C_2 = 477$ µM). Other compounds were defined as lipogenic; however, we did not include compounds AZ2–4 in the following multivariate analysis of TG effects due to sparse data. AZ5 was excluded from compound ranking as its systemic exposure was too low to correctly compare its lipogenic activity with that of other ligands.

**HDX-MS analysis of LXR modulators**. To follow conformational perturbations of LXRs upon binding of each studied ligand, we used differential HDX-MS where each isoform in its apo state was used as a reference. Based on an initial full HDX-MS kinetic analysis of the LXRα and LXRβ LBDs in complex with AZ2, AZ7, AZ876, and T0901317 (Supplementary Fig. 6), 30- and 600-s labeling times capturing short- and long-term stabilization effects of ligand binding were selected for screening. A differential HDX-MS analysis was then performed for a full set of ligands (Fig. 2, Supplementary Table 1, Supplementary Figs. 7 and 8). Good sequence coverage was achieved for both isoforms (Supplementary Fig. 9) resulting in a high spatial resolution of obtained data.

Overall, most of the selected compounds displayed similar differential HDX patterns with a strong protection from deuterium exchange observed for the peptides lining the LBP: central part of H3, H5, β-sheet, and helices H6, H7, and H10/11. LXRβ was better stabilized by most of the tested compounds in comparison with LXRα. Helices H3, H5, and H7 of LXRβ were stabilized by all compounds, with the strongest protection ranging from 30 to 60% observed for H3 after 600 s of deuterium exchange. Interestingly, the corresponding region of H3 in LXRα displayed a much broader range of ligand-dependent protection after 600 s: from 60% in BMS-852927 complex to no effect in the case of compounds LXR-623, AZ5, AZ8, and AZ9. Intriguingly, GW3965, and to a lesser extent, WAY-254011 destabilized both LXRα and LXRβ LBDs in the distal regions of C-terminal H3/H4, loop H8–H9, N-terminal H9, and in LXRβ additionally H1 (Supplementary Figs. 7 and 8).

Focusing on the AF-2 of LXRs, a short-term ligand stabilization effect on this region was observed. LXRβ H12 and the C-terminal H11 showed an average 17% protection after 30 s of deuterium exchange by all ligands except BMS-852927. Residual stabilization was observed for LXRβ H11 after 600 s of exchange in a ligand-dependent manner (5–14% protection). Ligand-dependent stabilization of LXRα H11–H12 was only observed after 30 s of labeling, suggesting that this region is more dynamic in LXRα compared with LXRβ. Compounds AZ5, 24,25EC, BMS-852927, and F1 did not stabilize LXRα H12 even though two of them (AZ5 and F1) displayed strong agonistic properties in the Gal4 LXRα reporter gene assay. 24,25EC did not exhibit agonistic activity in Gal4 assays; however, it was still able to stabilize LXRβ H12 in solution. Another inactive compound in the reporter gene assay was BMS-852927, and it is clearly distinct from other LXR ligands in the

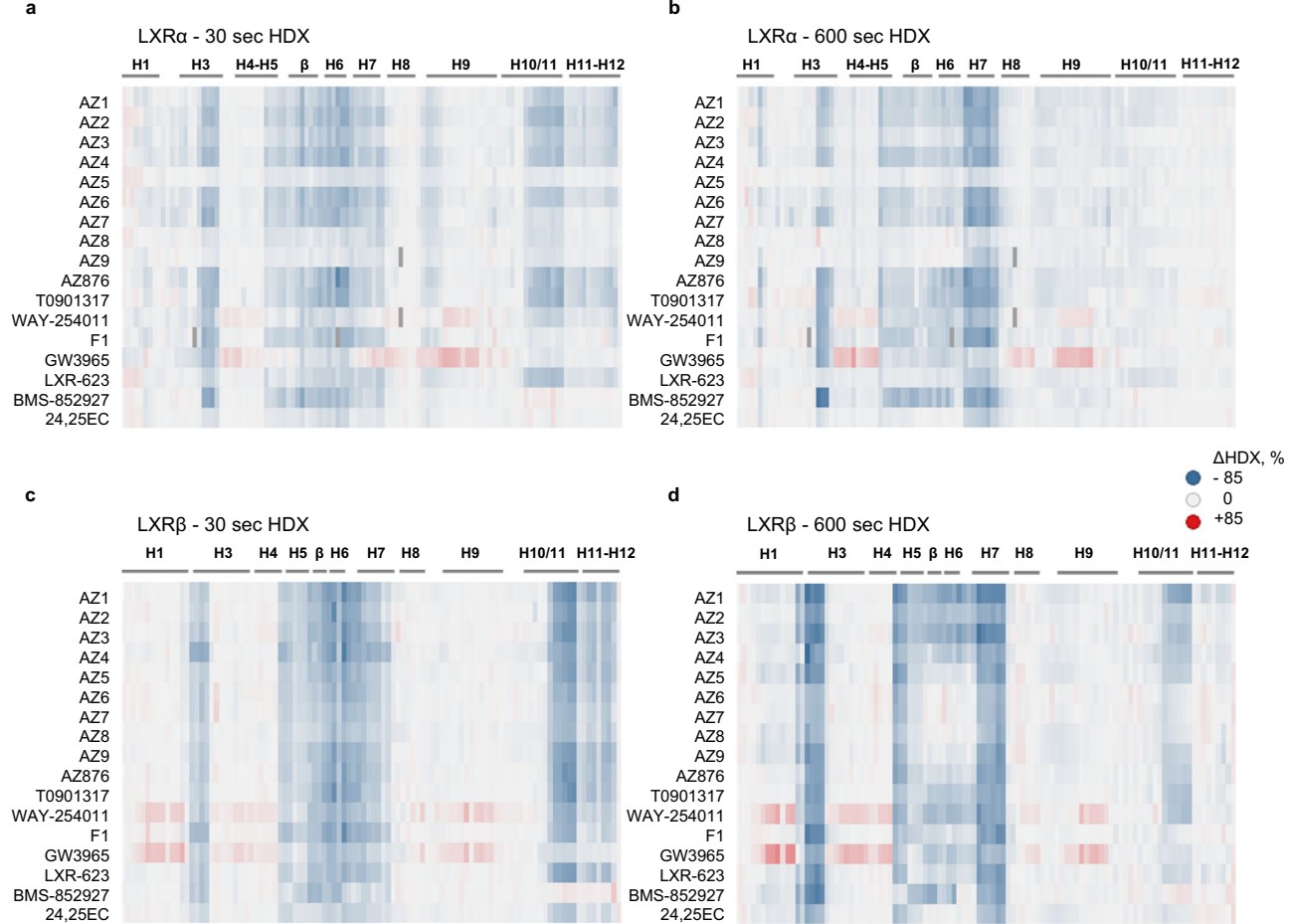

**Fig. 2** LXRα and LXRβ are differently stabilized by ligands. Fractional uptake difference heatmaps showing the variations of deuterium uptake between apo- and ligand-bound LXRα (**a**, **b**) and LXRβ (**c**, **d**) LBD peptides. Each row represents a compound and each column represents a peptide. A negative difference value (blue) corresponds to protection and a positive difference value (red) to deprotection from deuterium uptake upon ligand binding; large absolute values correspond to a strong effect. Differences in deuteration levels are shown for two time points: 30 s (**a**, **c**) and 600 s (**b**, **d**). Deuterium uptake values for all peptides were corrected to back exchange by using the fully deuterated controls. Missing peptides are shown in dark gray. Secondary structure is indicated on top of the heatmap. HDX-MS measurements were performed in triplicates. The complete peptide list and numerical values of differences in deuteration between apo- and ligand-bound states are shown in Supplementary Table 1.

HDX-MS analysis. While this compound strongly stabilizes LBP, particularly H3 region, it does not protect H11–H12 and therefore does not appear to favor what is considered to be a transcriptionally active receptor conformation (Supplementary Figs. 7 and 8). Another weak stabilizer of LXRβ H12 was GW3965. Analysis of crystal structures of LXRβ LBD in complex with WAY-254011, AZ3, AZ6, and AZ8 (Supplementary Fig. 10, Supplementary Table 2), as well as of previously published crystal structures of LXRβ[31,40,55] reveals that GW3965 and BMS-852927 are the only compounds not forming a direct hydrogen bond with LXRβ His435 (Supplementary Fig. 11) and therefore they do not stabilize the His435–Trp457 activation switch crucial for the agonistic conformation of H12. However, in the case of GW3965, its chlorotrifluoromethylbenzyl group is involved in multiple hydrophobic interactions with Val439, Leu442, Leu449, Leu453, and Trp457 that partially compensate for the hydrogen bonding loss (Supplementary Fig. 11i). For the BMS-852927, only two out of four LXRβ molecules present in the asymmetric unit contain H12 in its active conformation, and in those complexes, H11 is shifted away from the ligand by 3.5 Å, which leads to the loss or weakening of the interactions with the C terminus (Supplementary Fig. 11j, k). In the 24,25EC/LXRβ complex, His435 is in a suboptimal position at a

distance of 3.4 Å from the epoxide of the sterol side chain and thus it forms a weak hydrogen bond with the ligand. Other agonists form shorter hydrogen bonds of 2.5–3.1 Å essential for proper position of H12, which correlates with their activities both in LXRβ HDX-MS and functional assays. AF-2 of LXRα was more differentially protected in the HDX-MS experiments. However, among the compounds used in this study, only structures with T0901317[42] and GW3965[32] have been reported, which makes it difficult to compare HDX-MS signatures with high-resolution structures. LXRα H12 is stabilized by T0901317 but not by GW3965 via direct hydrogen bonds to His421 (corresponding to LXRβ His435), which is also reflected in the HDX-MS measurements. In summary, the crystallographic information supports the HDX-MS data with respect to stabilization of H12.

Taken together, our HDX-MS data demonstrate that compounds with different chemical structures protect overlapping but nonidentical regions of the LXRα and LXRβ LBDs, and the degree of protection from solvent exchange varies between ligands. To investigate whether such differences in HDX patterns could serve as fingerprints for pharmacological effects of ligands, we transformed the HDX-MS data into a matrix of variables (predictors) and applied a multivariate statistical analysis strategy

to separate conformational changes contributing to lipogenesis from those associated with the induction of *Abca1* in the intestine.

**Increased plasma TG levels correlate to ligand-induced active conformation of LXRs.** HDX-MS data collected for 115 and 103 unique peptides identified in LXRα and LXRβ, respectively, across 17 compounds were used for building a multidimensional data matrix. An unsupervised principal component analysis (PCA) of the data showed that lipogenic ($n = 5$) and non-lipogenic ($n = 7$) compounds separated along the first principal component (Fig. 3a), confirming that the HDX-MS dataset contains variable trends driving separation of compounds according to their lipogenicity ranking. Among compounds not assigned to any TG class, AZ5 and 24,25EC clustered together with compounds with low lipogenic activity, while AZ2–4 clustered together with lipogenic compounds. Supervised orthogonal partial least-squares discriminant analysis (OPLS-DA) and the associated S plot were subsequently used to identify peptides driving the separation between lipogenic and non-lipogenic compounds as well as their relative contributions (Fig. 3b, Supplementary Fig. 12 and Supplementary Table 3). Identified peptides cover similar regions of LXRα and LXRβ comprising the central part of H5, helices H6, H7, H10/11, and H12 (Fig. 3c). A stronger protection from deuterium exchange by lipogenic ligands compared with non-lipogenic compounds was observed for all discriminating peptides (Fig. 3d, e). In agreement with crystal structures of LXRα and LXRβ, the indicated protein regions largely contribute to the agonist-induced formation of the AF-2. However, the difference in stabilization of the helix H12 by different ligand classes was not statistically significant in the case of LXRβ.

**Lipogenic ligands enhance LXRα interaction with the SRC1 coactivator peptide.** To investigate whether an increased stabilization of the active states of LXRα and LXRβ by lipogenic compounds triggers the formation of the coactivator-binding surface and causes differential coactivator recruitment, we studied ligand-induced interactions between LXRs and the peptide comprising the second receptor-interacting motif of the coactivator SRC1. First, we measured the binding of varying concentrations of receptors in the presence of saturating amounts of ligands to the immobilized SRC1 peptide by surface plasmon resonance (SPR). Unliganded LXRα and LXRβ did not interact with the coactivator peptide under the conditions tested (Supplementary Figs. 13 and 14). All compounds induced binding of LXRs to the SRC1 peptide (Fig. 3f); however, different trends were observed for the two isoforms. LXRα displayed a wide range of binding affinities to the SRC1 peptide, with clear discrimination between lipogenic compounds strongly enhancing the interaction (average Kd = 1.2 μM) and non-lipogenic compounds acting as weak agonists and inducing modest LXRα affinity to the SRC1 peptide (Kd above 4.5 μM), apart from AZ6 (Kd = 1.5 μM). A much stronger ligand-induced interaction between LXRβ and the coactivator peptide was observed, with a smaller variation between LXRβ agonists (Kd for most of compounds below 2 μM). 24,25EC induced modest coactivator binding (Kd = 2.5 μM), and BMS-852927 promoted only a weak interaction with SRC1 (Kd = 19.8 μM) in agreement with the HDX-MS data showing that this compound prevents H12 from folding into an active conformation. Stronger interaction of LXRβ with the coactivator peptide is also supported by the HDX-MS data showing that LXRβ is more stabilized by ligand binding, particularly in the regions of H10/11 and H12. LXRβ adopts an active conformation necessary for the coactivator recruitment more readily and this can be induced even by weak agonists.

We next collected HDX-MS data for 17 ligand-bound LXR complexes in the presence of saturating amounts of the SRC1 peptide. In agreement with the binding data, addition of the coactivator peptide to agonist-bound LXRs led to a large stabilization effect on the activation helix H12 and helices H3, H5, and H10/11 of both receptor isoforms, exceeding the ligand-only-induced protection from exchange by 20–30% (Supplementary Fig. 15 and Supplementary Table 4). Multivariate analysis of the LXR-coactivator HDX-MS dataset resulted in an improved OPLS-DA TG model (Supplementary Fig. 16) and identified solely LXRα peptides as contributing to lipogenic class separation, contrary to the initial model where predictive variables included peptides from both LXRα and LXRβ. Significant peptides comprised regions of LXRα H3, H6–H7, H10/11, and H12 (Fig. 3g and Supplementary Table 5). While all compounds were able to induce coactivator peptide recruitment leading to a long-term stabilization of H12, lipogenic compounds were better stabilizers of the LXRα/SRC1 complex ensuring a better protection of the H12 (Fig. 3h).

Taken together, our data suggest that lipogenic compounds appear to be more agonistic to LXRα than compounds inducing lower levels of plasma TG. This finding correlates with previous studies showing that LXRα highly expressed in the liver is the primary isoform responsible for the undesirable lipogenic effects of LXR agonists[23,28].

**Conformational fingerprints of intestinal Abca1 induction.** The unsupervised PCA of the HDX-MS data showed that ligands inducing high ($n = 12$) and low ($n = 4$) levels of the intestinal *Abca1* mRNA separated along the second principal component (Fig. 4a). In contrast, lipogenic and non-lipogenic compounds separated along the first principal component (Fig. 3a), indicating that the HDX-MS data contain internal trends of predictors for the *Abca1* induction distinct from those of the TG effects. The OPLS-DA model (Supplementary Fig. 17) and the associated S plot identified 33 LXRα peptides with the highest contribution to the separation between two ABCA1 classes (Fig. 4b and Supplementary Table 6). No LXRβ peptides passed the significance selection criteria, suggesting that the induction of intestinal *Abca1* transcript levels largely correlates to the compound ability to stabilize LXRα. Peptides from the central part of H3 and the region comprising the C terminus of H5, β-sheet, and H6 displayed the highest contribution to the ABCA1 class separation; other significant peptides derived from the central parts of H5 and H7 (Fig. 4c, d and Supplementary Table 6). Similarly to the TG effects, compounds inducing higher functional response displayed stronger stabilization of the contributing peptides (Fig. 4d). Interestingly, the region of LXRα H3 showing the largest difference in deuteration between high and low ABCA1 inducers appears to be more dynamic than the corresponding portion of H3 from LXRβ with more than 20% higher deuterium uptake after 30–600 s of solvent exchange (Fig. 4e). Different dynamics of this region could be important for differential cofactor recruitment by LXR isoforms as the central portions of H3 and H5 are known to form a hydrophobic groove on the surface of the LBD where both coactivators and corepressors of NHRs bind (Supplementary Fig. 18).

**Ligand-induced release of the NCOR1 peptide does not correlate to the ABCA1 activity of compounds.** Since the reporter gene assay typically used for screening of agonists did not predict ligand ability to induce intestinal *Abca1* (Supplementary Fig. 19a), and compounds inducing high levels of *Abca1* mRNA did not enhance LXR affinity to the coactivator peptide over low inducers (Supplementary Fig. 19b), we investigated whether ligand-dependent

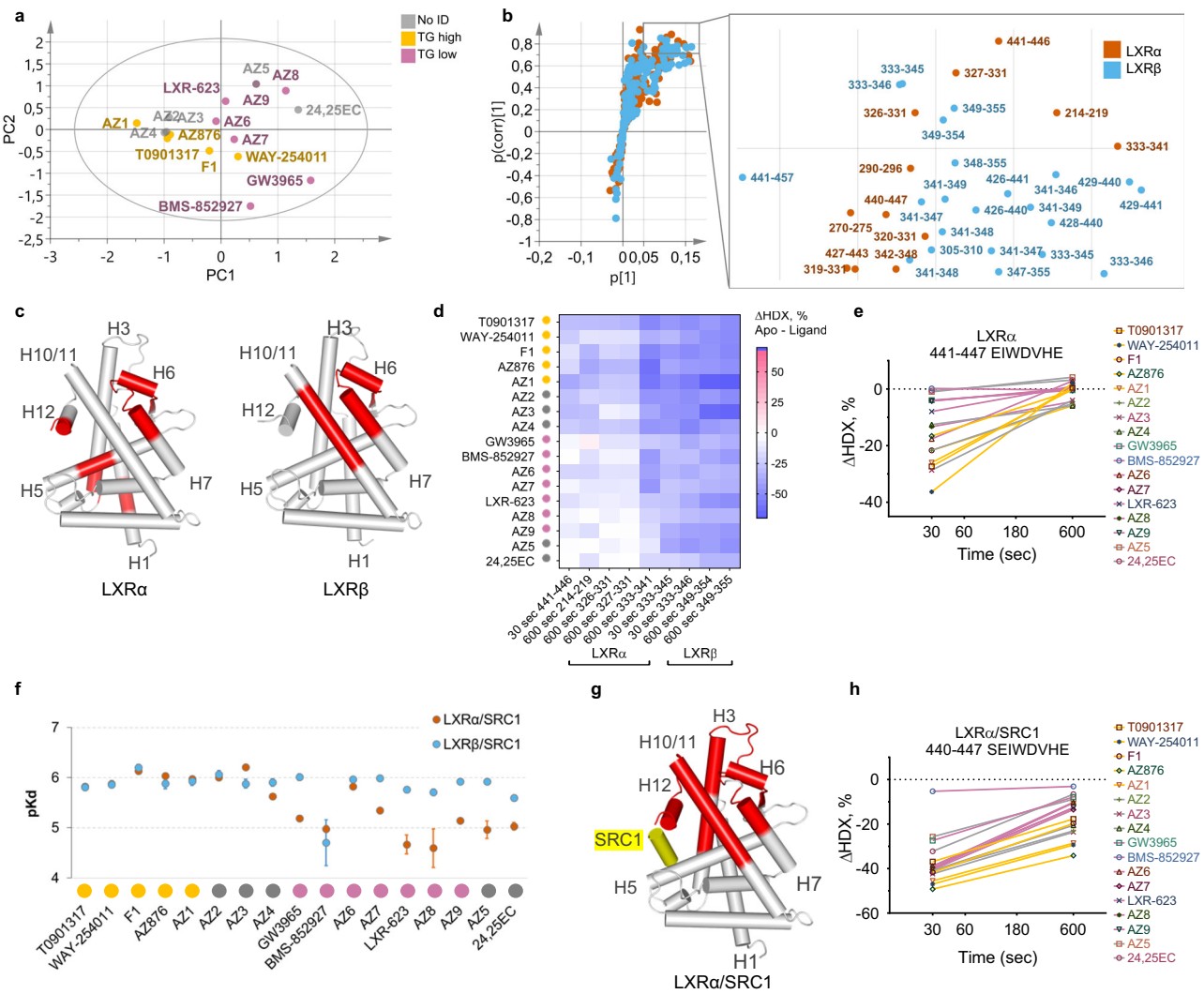

**Fig. 3** LXR ligands inducing high plasma TG stabilize active receptor conformation. **a** PCA score scatter plot of the first two principal components. Yellow dots: lipogenic LXR ligands, purple dots: non-lipogenic LXR ligands, and gray dots: compounds with no assigned TG class. $R^2X(cum) = 83.4\%$, $Q^2(cum) = 62.9\%$, $n = 17$. **b** S plot of the TG OPLS-DA model and the zoomed-in part of the plot representing limits used for the selection of differential peptides with the highest contribution to class separation ($|p(corr)| > 0.7$, $p > 0.05$). Orange dots: LXRα peptides, cyan dots: LXRβ peptides. **c** Statistically significant ($p$-value < 0.05, two-tailed Student's $t$ test) predictive peptides mapped on the LXRα (2ACL.pdb) and LXRβ (1PQC.pdb) crystal structures. **d** Average differences in deuteration levels of the differential peptides having the highest correlation to the class separation ($|p(corr)| > 0.8$). **e** Differential deuterium uptake levels of the LXRα peptide 441–447 in the presence of lipogenic (yellow) and non-lipogenic (purple) compounds. Ligands not assigned to any TG class are shown in gray. **f** Binding affinities of LXRα (orange circles) and LXRβ (cyan circles) to the immobilized coactivator peptide measured by SPR (±standard error). TG compound classes are shown below for clarity: lipogenic ligands in yellow, non-lipogenic in purple, and compounds with no assigned TG class in gray. Original sensorgrams and derived equilibrium-binding affinity parameters are shown in Supplementary Fig. 13 (LXRα) and Supplementary Fig. 14 (LXRβ). **g** Statistically significant ($p$-value < 0.05, two-tailed Student's $t$ test) predictive peptides mapped on the LXRα crystal structure (2ACL.pdb). **h** Differential deuterium uptake levels of the LXRα peptide 441–446 in the presence of lipogenic (yellow) and non-lipogenic (purple) compounds and the SRC1 coactivator peptide. Ligands not assigned to any TG class are shown in gray. HDX-MS data were collected in triplicate, and deuteration data are normalized to the fully deuterated control (±standard deviation).

ABCA1 effects observed in vivo could originate from the dissociation of the LXR-corepressor complexes by measuring the interactions between LXRs and the NCOR1 corepressor peptide comprising the second NHR recognition motif.

First, we confirmed the binding mode of the corepressor peptide to the apo states of LXRα and LXRβ by using HDX-MS. Similar regions of LXRα and LXRβ were affected by the NCOR1 binding with the highest protection observed for the central parts of H3 and H5 forming the corepressor-binding groove (Fig. 5a, b). Average reduction in deuterium uptake across the five labeling points was 14–17% for both helices (Fig. 5a and Supplementary Table 7). Of note, the NCOR1-interacting region of H3 comprises

the same region as identified by the OPLS-DA ABCA1 model (LXRα residues 260–269).

Next, we assessed the ability of ligands to inhibit binding of LXRs to the immobilized NCOR1 peptide by SPR. Both isoforms in apo states bind to the corepressor peptide (Kd LXRα/NCOR1 = 0.8 µM, Kd LXRβ/NCOR1 = 0.5 µM) (Supplementary Figs. 20 and 21). Interestingly none of the compounds inhibit the LXRα/NCOR1 interaction. In contrast, the LXRβ/NCOR1 interaction is inhibited with a wide range of potencies (Fig. 5c) and is completely blocked by the compounds AZ1–4 and WAY-254011. BMS-852927 enhanced the interaction of both LXRα and LXRβ with the corepressor, hence acting as an

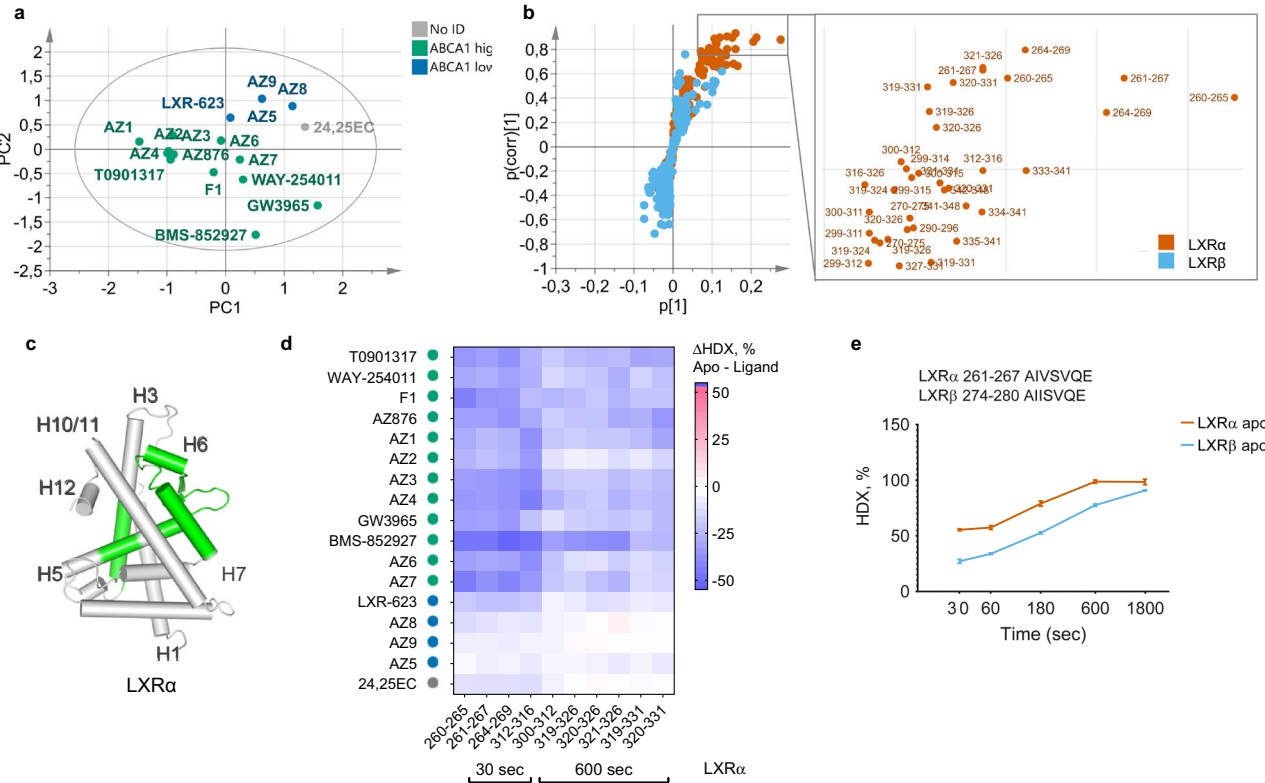

**Fig. 4** Multivariate analysis reveals structural determinants of high intestine *Abca1* induction. **a** PCA score scatter plot of the first two principal components. Green dots: LXR ligands inducing high intestine *Abca1* levels, blue dots: low inducers of ABCA1, and gray dots: compounds with no assigned ABCA1 class. $R^2X(cum) = 83.4\%$, $Q^2(cum) = 62.9\%$, $n = 17$. **b** S plot of the ABCA1 OPLS-DA and the zoomed-in part of the plot representing limits used for the selection of differential peptides with the highest contribution to class separation ($|p(corr)| > 0.7$, $p > 0.05$). Orange dots: LXRα peptides, cyan dots: LXRβ peptides. **c** Statistically significant ($p$-value $< 0.05$, two-tailed Student's $t$ test) predictive peptides better stabilized by potent ABCA1 inducers mapped on the LXRα crystal structure (2ACL.pdb). **d** Average differences in deuteration levels of the differential peptides having the highest correlation to the class separation ($|p(corr)| > 0.8$). **e** Time-point deuteration kinetics of the central part of H3 (LXRα peptides 261–267, orange and LXRβ peptides 274–280, cyan). HDX-MS data were collected in triplicate (±standard deviation), and deuteration data are normalized to the fully deuterated control.

inverse agonist in this assay. No clear correlation was observed between the intestinal *Abca1* induction and the NCOR1 peptide dissociation.

**Uncoupling beneficial and adverse effects of LXR ligands**. The goal of this study was to identify unique receptor regions that strongly correlate with the induction of *Abca1* by LXR ligands but minimally contribute to increase in TGs. We compared the two models (ABCA1 and TG) by using a Shared and Unique Structures (SUS) plot, in which the correlation coefficients of HDX-MS peptides for the two models are plotted against each other to reveal differences and similarities in peptide contributions (Fig. 6a). In the SUS plot, a high p(corr) indicates a high prominence of the peptide in driving the separation between groups in either model. The peptides that are of equally high importance for the two models cluster along the diagonal. Peptides strongly contributing to plasma TG but not *Abca1* induction cluster along the x axis. These peptides comprise H10/11 and H12 peptides of LXRs (Fig. 6b). Peptides correlating only to high *Abca1* induction are located along the y axis; they comprise H3 and partial H5–β-sheet region of LXRα. Targeting these regions of LXRα while avoiding interactions with H12 could serve as potential directions for development of ligands with improved therapeutic profiles. Based on the structural information, an apparent solution for weakening interactions with H12 would be preventing formation of the hydrogen bond with the LXRα His421 (LXRβ His435).

We also compared the TG and ABCA1 models with the OPLS-DA models obtained for the in vitro transcriptional activities of ligands in LXR-dependent reporter gene assays typically used for screening of LXR agonists (Supplementary Figs. 22 and 23). As expected, highly potent compounds strongly stabilized agonist conformation of the AF-2. Peptides contributing to the in vitro LXR transcriptional activities are largely shared with the TG but not with the ABCA1 model, suggesting that screening of compounds with reduced lipogenic activities could benefit from selection of ligands displaying modest potencies in the in vitro assays.

## Discussion

While the most common strategy for discovery of LXR modulators with potential therapeutic effects and minimal hepatic lipogenesis continues to be the development of LXRβ-selective compounds[33,37,41], in this study, we have demonstrated that the improved therapeutic profile of LXR ligands observed in phenotypic screening in mice largely depends on their ability to differentially stabilize the conformation of LXRα. We show that plasma TG induction likely results from agonist-induced stabilization of the LXRα AF-2 domain, whereas intestinal *Abca1* transcript levels are highly correlated with the stabilization of the peptides comprising the central part of the LXRα H3, H5–H7, and β-sheet. Our finding that the intestinal *Abca1* is upregulated by ligands that do not induce an active conformation of the H12 is intriguing and is best illustrated by BMS-852927 that is a potent inducer of *Abca1*

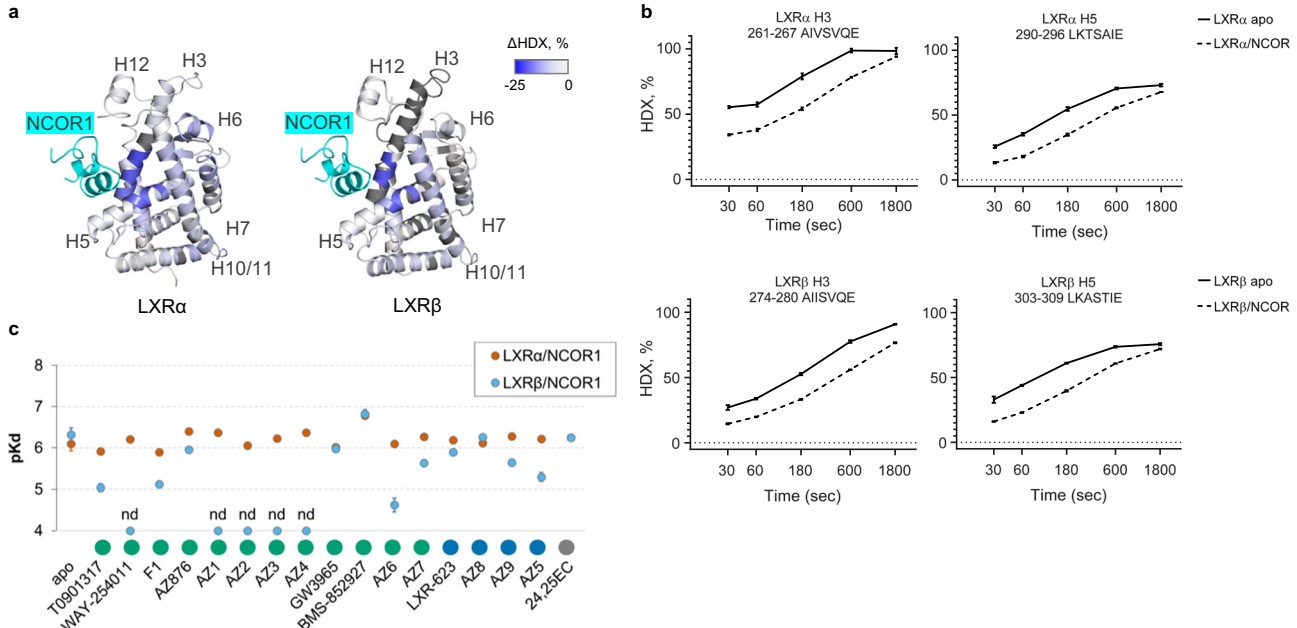

**Fig. 5** Functional implications of the LXR helices H3 and H5 in the cofactor recruitment. **a** Average differential HDX data mapped onto the LXRα (2ACL. pdb) and LXRβ (1PQC.pdb) crystal structures showing the difference in deuterium uptake between apo- and NCOR1-bound states. Full peptide list and statistical summary are presented in Supplementary Table 7. **b** Deuterium uptake plots for peptides from LXRα and LXRβ helices H3 and H5 in apo- and NCOR1-bound states. HDX-MS measurements were performed in triplicates; data are shown ± standard deviation. **c** Binding affinities of apo- and ligand-bound LXRα (orange circles) and LXRβ (cyan circles) to the immobilized NCOR1 corepressor peptide measured by SPR. Bars: ± standard error; nd: not detected. Original sensorgrams and derived equilibrium-binding affinity parameters are shown in Supplementary Fig. 20 (LXRα) and Supplementary Fig. 21 (LXRβ).

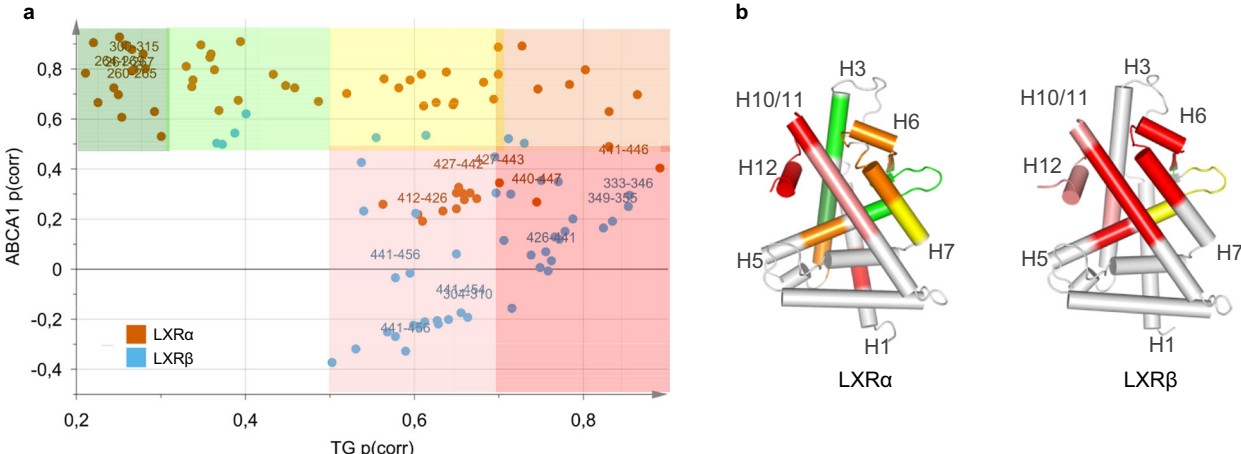

**Fig. 6** SUS analysis of TG and ABCA1 OPLS-DA models identifies potential directions for development of ligands with improved therapeutic profiles. SUS plot (**a**) and the corresponding p(corr) values mapped on the crystal structures of LXRα (2ACL.pdb) and LXRβ (1PQC.pdb) (**b**) according to shared color code. Dark green: p(corr) ABCA1 > 0.5, p(corr) TG < 0.3; light green: p(corr) ABCA1 > 0.5, 0.3 < p(corr) TG < 0.5; yellow: p(corr) ABCA1 > 0.5, 0.5 < p (corr) TG < 0.7; orange: p(corr) ABCA1 > 0.5, p(corr) TG > 0.7; red: p(corr) ABCA1 < 0.5, p(corr) TG > 0.7; pink: p(corr) ABCA1 < 0.5, 0.5 < p(corr) TG < 0.7. For clarity, peptides with low contribution to the class separation for both models (p < 0.05, p(corr) ABCA1 < 0.5, p(corr) TG < 0.5) were excluded. A complete list of p(corr) values for both models is provided in Supplementary Table 8.

in vivo but behaves as an antagonist in our HDX-MS study and as an inverse agonist in the LXR/NCOR1 interaction assays. BMS-852927 was initially characterized as an LXRβ-selective partial agonist[40] showing a favorable profile at efficacious doses in mice and cynomolgus monkeys. However, increased lipid levels were observed upon multiple dosing in phase 1 clinical studies[41]. Although our results confirm that BMS-852927 is non-lipogenic when tested in preclinical murine models, we also observe that BMS-852927 can promote complex formation between LXRα and

SRC1. These results suggest that residual H12-dependent coactivator recruitment could be crucial in the liver where LXRα and coactivators are present at high levels.

We identified the central portions of LXRα H3 and H5 as receptor regions preferentially stabilized by compounds inducing high transcript levels of intestinal *Abca1*. This region forms interaction surfaces for both coactivators and corepressors. LXRs actively recruit corepressors NCOR1 and SMRT to repress basal expression of target genes[56], and dissociation of NCOR1 and

SMRT was previously shown to result in derepression of the *Abca1*, but not of the *Srebp1c* gene in macrophages[4]. Since stabilization of H12 does not appear to be required for *Abca1* upregulation, it could be hypothesized that ligand-dependent changes in the conformation of LXRα H3–H5 could result in the dissociation of corepressor complexes leading to increased ABCA1 expression. Although we could not detect differences in inhibition of NCOR1 peptide binding to the LXR LBDs when compounds from different classes of ligands were tested, our findings suggest that future work investigating LXR/corepressor complexes and how they are modulated by non-H12-stabilizing ligands will be required. Schulman et al. demonstrated that it is possible to detect ligand-induced H12-independent interactions between RXRs and components of the TFIID complex (TAF4)[57], and this association is strongly promoted by partial RXR agonists[58]. Non-H12-dependent mechanisms of action associated not only with changes in coregulator recruitment but also with the altered post-translational modification status of the receptor have been described for PPARγ partial agonists[59–61].

The HDX-MS analysis revealed several important dissimilarities in structural dynamics of LXRα and LXRβ that could have major implications for the observed functional differences between two receptor isoforms. First, AF-2 of LXRα is less stabilized by agonist ligands and forms weaker complexes with the coactivator peptide compared with LXRβ. This indicates that LXRβ adopts an agonist conformation more easily than LXRα, by explaining identification of a great number of compounds that appear to selectively activate LXRβ in vitro despite the LBPs of LXRα and LXRβ being almost identical. Second, the central region of LXRα H3 highly correlated to *Abca1* induction is more flexible in apo LXRα compared with LXRβ. Distinct conformational flexibility of such a highly reactive region is likely to affect the cofactor recruitment. Interestingly, similar observations that helix H3 dynamics differs between two receptor isoforms and can potentially be a regulatory switch were previously reported for ERα and ERβ[62].

Development of partial agonists as therapeutics has proven to be a viable strategy for NHRs, including PPARα and PPARγ, as weak inducers of H12-dependent coactivator interactions often display a dissociation between positive and adverse effects. Our data suggest that similar approach could be applied for a rational development of selective LXR modulators. We used multivariate analysis to demonstrate that in vitro transactivation assays are largely H12-dependent, supporting their common use for the selection of potentially non-lipogenic weak LXRα agonists[33,40]. On the other hand, reporter gene assay would not be a method of choice when less- or non-H12-dependent transcriptional responses of NHRs are pursued. Here we clearly demonstrate how drug development could benefit from the HDX-MS, a pseudostructural biophysical method providing receptor–ligand interaction details at the peptide level. A combination of HDX-MS and multivariate analysis presents an attractive quantitative approach that can be used to identify desired conformational fingerprints of ligands. In addition to previously shown complementarity of HDX-MS and in vitro assays for drug discovery[46–48], here we demonstrate for the first time, to the best of our knowledge, that HDX-MS can also be coupled to in vivo phenotypic screening, there by providing a powerful tool for the identification of compounds with optimized therapeutic profiles.

## Methods
**Compound synthesis and characterization.** Compound synthesis and characterization data are reported in Supplementary Note 1: synthetic procedures.

**Expression and purification of LXRα and LXRβ LBDs.** Human his-tagged LXRα LBD (200-447) cloned into pET28 (Novagen, Gibbstown, NJ, USA) and human

his-tagged LXRβ LBD (216-461) cloned into pET24D were expressed in *Escherichia coli* BL21(DE3) by batch fermentation. Proteins were purified by chelating sepharose FF loaded with Ni$^{2+}$ followed by Superdex 75 size-exclusion chromatography. The storage protein buffer was 20 mM Bis-Tris Propane (pH 9.0), 150 mM NaCl, 10% glycerol, and 1 mM TCEP.

**Binding and transactivation assays.** Binding affinities were determined in the displacement SPA (Scintillation Proximity Assay) assays by using recombinant LXRα and LXRβ LBDs and ten concentrations of ligands competing with the binding of [$^3$H]-T0901317. LXRα (0.5 μg) or LXRβ (0.25 μg) proteins were mixed with 0.1 mg of the SPA beads (polylysine-coated yttrium silicate beads, RPNQ0010P, GE Healthcare, Piscataway, NJ, USA), 30 nM [3 H]-T0901317, and test compound in assay buffer (20 mM Tris, pH 7.5, 80 mM NaCl, 2 mM dithiothreitol, 0.125% Chaps, and 10% glycerol). The assay mixture was shaken gently for 2 h in a Wallac Isoplate 1450-514 on a plate shaker followed by 1-h incubation to allow the beads to settle before counting.

Transactivation assays were performed in U2OS osteosarcoma cells (ATCC, USA). About 2.5 μg of the expression vectors pSG5-Gal4-LXRα LBD or pSG5-Gal4-LXRβ LBD (Stratagene, La Jolla, CA, USA) were co-transfected with 25 μg of a pGL3 luciferase reporter plasmid containing a minimal SV40 promoter (Promega, Madison, WI, USA) and five copies of the UAS Gal4 recognition site and 22.5 μg of pBluescript (Stratagene) via electroporation. Ligands were added at ten different concentrations 2 h post transfection and the luciferase activity was measured after 40 h.

The compounds were run in triplicate (binding assay) or in quadruplicate (transactivation assay). Standard deviations are indicated for individual experiments run more than once.

**Animals.** All procedures involving animals were conducted in accordance with accepted standards of good animal practice and approved by the respective local animal ethics committees (Gothenburg region Local Ethics Review Committee on Animal experiments and University of Virginia Animal Care and Use Committee). Male C57BL/6JOlaHsd mice were retrieved from Harlan Laboratories (Horst, The Netherlands) around 7 weeks of age. In the studies with BMS-852907, male 8-week C57BL/6J mice (Jackson Laboratories) were used. Mice were given ad libitum access to water and standard rodent chow during the studies.

**Compound formulation.** Solid compounds were solubilized in PEG400/Ethanol/Solutol/Water solution (36.3/9.1/4.6/50%). Compounds were given daily by oral gavage, 5 ml kg$^{-1}$, at doses ranging from 0.1 to 100 μmol kg$^{-1}$ of body weight for 4 days. Animals were terminated 3 h after the last dose.

**Termination of the experiment.** At termination, the whole-blood volume was drawn from isoflurane-anesthetized mice via an incision in the carotid artery into EDTA blood tubes (S-Monovette$^®$, EDTA KE/1.3 mL, Sarstedt, Nümbrecht, Germany or BD Microtainers$^®$, Benton, Dickinson and Company, Franklin Lakes, New Jersey, USA). Blood samples were placed on ice and centrifuged within an hour (15 min, 2500 × $g$, 4 °C or 5 min, 16000 × $g$, 4 °C). The plasma was then dispensed in separate tubes and kept at −80 °C for later analysis of compound concentration and TG levels. For the analysis of intestinal *Abca1* transcript levels, a 3–5-mm long segment of the duodenum was excised and cleared from any gut contents. Tissues were instantly snap frozen in liquid nitrogen and stored at −80 °C.

**RNA preparation and qPCR.** Snap-frozen intestinal tissues were quickly disrupted in QIAzol$^®$ (Qiagen, Hilden, Germany) or PureZOL$^™$ (Bio-Rad Laboratories, Hercules, California, USA) by using a bead tissue homogenizer. Lysates were mixed with chloroform and centrifuged at 6000 × $g$ for 15 min at 4 °C. Total RNA was purified from the upper aqueous phase on the ABI PRISM$^™$ 6100 Nucleic Acid PrepStation (Applied Biosystems, Foster City, USA) or by using the RNeasy kit$^®$ (QIAGEN Sciences Inc., Germantown, Maryland, USA) including on-column DNase treatment by using the respective manufacturer's protocol and reagents. Concentration of total RNA was determined on a NanoDrop$^®$ ND-100 Spectrophotometer (Saveen Werner, Lögstrup, Sweden). The RNA templates were transferred to cDNA by reverse transcription by using the High-Capacity cDNA Reverse Transcription Kit (Applied Biosystems, Foster City, USA) according to the manufacturer's protocol. Mouse ATP-binding cassette subfamily A member 1 (Abca1) was analyzed by using the forward primer 5′-AAGGGTTTCTTTGCTCAGATT GTC-3′, reverse primer 5′-TGCCAAAGGGTGGCACA-3′, and the dual-labeled probe 5′-6FAM-CCAGCTGTCTTTGTTTGCATTGCCC-TAMRA-3′. Mouse ribosomal protein large P0 (Rplp0, 36b4) was analyzed by using the forward primer 5′-GAGGAATCAGATGAGGATATGGGA-3′, reverse primer 5′-AAGCAGGCTG ACTTGGTTGC-3′, and the FAM-labeled probe 5′-TCGGTCTCTTCGACTAAT CCCGCCAA-TAMRA-3′. In the study with BMS-852907, RT-qPCR reactions were carried out in separate triplicate wells for Abca1 and Rplp0, respectively, by using 20 ng of cDNA template, 385 nM of the above-mentioned primers, and iQ$^™$ SYBR Green Supermix (Bio-Rad, Hercules, California, USA) on a Bio-Rad MyiQ instrument (Bio-Rad, Hercules, California, USA). In the other studies, 400 nM Abca1 forward and reverse primers, 200 nM Abca1 probe, 200 nM Rplp0 forward and reverse primers, and 100 nM Rplp0 probe were mixed with 15 ng of cDNA

template and TaqMan™ Gene Expression Master Mix (Applied Biosystems, Foster City, USA). The RT-qPCR reactions were performed on a 7500 Real-time PCR system (Applied Biosystems, Foster City, USA). Cycle threshold values (Ct) were retrieved from the instrument software. Relative changes in mRNA expression were calculated by using the comparative cycle method ($2^{-\Delta\Delta Ct}$).

**Plasma triglycerides.** Plasma triglyceride levels were analyzed by using the Triglycerides/Glycerol Blanked reagent kit (Roche Diagnostics GMBH, Mannheim, Germany). Samples were measured on an ABX Pentra 400 instrument (Horiba Medical, Irvine, California, USA) or Cobas Mira plus instrument (Roche Diagnostics Corporation, Indianapolis, USA). Samples from the study with BMS-852907 were measured in triplicate by using a colorimetric assay (Pointe Scientific, Inc., Canton, Michigan, USA) formatted for 96-well plates. Absorbance was measured by using a Tecan Infinite M200 plate reader (Tecan Austria GmbH, Salzburg, Austria).

**Quantification of drug concentration in plasma.** About 25 µL of plasma was precipitated with 150 µL of cold acetonitrile containing 0.2% formic acid. After vortexing followed by 20 min of centrifugation at $3220 \times g$ and 4 °C, the supernatant was diluted 1:1 with 0.2% formic acid (aqueous phase). Sample analysis was performed by using a reversed-phase high-pressure liquid chromatography system coupled to a Quattro Ultima triple-quadruple mass spectrometer (Waters, UK). A rapid 2-min 4–90% (vol/vol) acetonitrile (containing 0.2% formic acid) gradient was used for chromatographic separation. The lowest limit of detection varied between compounds and generally was in a low nanomolar range.

**Modeling of in vivo data.** The relationship between administered dose and the observed drug-induced *Abca1* mRNA levels in the intestine was described by an ordinary $E_{max}$ model

$$E = BL + \frac{E_{max} \cdot Dose}{ED_{50} + Dose} \quad (1)$$

where BL is the baseline level of fold *Abca1* transcript induction when no drug is present (i.e., set to 1). To determine the potency ($ED_{50}$) of the different drugs, all data were simultaneously fit by using a population approach where the compounds were assumed to have the same maximal fold induction ($E_{max}$) and drug was added as a categorical covariate on $ED_{50}$. AZ6 was selected as the reference drug with which all the other compounds were compared.

For plasma TGs, the relationship between the observed drug concentration and the fold increases in TG concentration in plasma was described by a log-linear function

$$TG = BL + \ln(1 + m \cdot C^{\gamma}) \quad (2)$$

where BL is the fold TG level when no drug is present (i.e., 1), and $m$ and $\gamma$ are scaling factors influencing the amplitude and the slope of the drug concentration–TG relationship, respectively. Data for all drugs were simultaneously fit where the compounds were assumed to have the same $\gamma$ and drug was added as a categorical covariate on $m$. The compound T0901317 was selected as the reference compound. To rank the compounds, the drug concentration ($C_2$) needed to generate a twofold increase in plasma TG was estimated by rearranging Eq. (2):

$$C_2 = \left( \frac{\exp^{(2fold - BL)} - 1}{m} \right)^{\frac{1}{\gamma}} \quad (3)$$

All modeling was done in Phoenix NMLE 1.3 (Certara, L.P., 210 North Tucker Boulevard Suite 350, St. Louis, MO 63101, USA) by using the Naïve pool estimation method and either a multiplicative (*Abca1* mRNA) or additive (TG) normally distributed observation error model.

**Crystallization and structure determination.** Prior to co-crystallization, AZ6, AZ8, and WAY-254011 were incubated with the LXRβ LBD in 20 mM NH₄Cl, pH 9.5 (13 mg ml⁻¹) and the coactivator peptide SRC-2 (KHKILHRLLQDSS) overnight at 4 °C. AZ6 (2 mM in DMSO) was co-crystallized with LXRβ in the presence of 2 mM of the SRC-2 peptide in 0.1 M HEPES, pH 7.5 and 18% PEG6000. The data were collected at beamline ID29 at the European Synchrotron Radiation Facility (ESRF). AZ8 (5 mM in DMSO) was co-crystallized with LXRβ in the presence of 1 mM SRC-2 peptide in 100 mM PIPES, pH 7.0 and 17% PEG4000. The data were collected at beamline ID14-2 (ESRF). WAY-254011 (2 mM in DMSO) was co-crystallized with LXRβ in the presence of 2 mM of the SRC-2 peptide in 100 mM Bis-Tris, pH 6.5, 2 M sodium formate, and 100 mM NaCl. The data were collected at beamline ID29 (ESRF).

The data sets were processed by using Mosflm[63] and Scala[64]. Structures were solved by molecular replacement by using programs from the CCP4 suite[65]. Refinement was performed by manual rebuilding in Coot[66] and automatic refinement by using Refmac5[65] or CNS[67]. The ligands were added to the models toward the end of the refinement. For the AZ6 complex, the asymmetric unit contained four LBD complexes and three additional copies of the ligands were modeled in the crystal interfaces. For the AZ8 and WAY-254011 complexes, the asymmetric unit contained one LBD complex. One additional copy of WAY-254011 was modeled in the crystal interface. Statistics on data collection and refinement are available in Supplementary Table 2.

The structure of LXRβ in complex with AZ3 was purchased from Proteros biostructures GmbH.

**HDX-MS.** Prior to the deuterium incorporation experiments, 10 µM apo LXR was incubated with 10× excess of ligand or equal volume of DMSO for 30 min at room temperature. In the experiments with cofactors, 15× excess of the SRC1 coactivator peptide and the NCOR1 corepressor peptide were added to the ligand-bound or apo proteins, respectively, and incubated for 30 min at room temperature. Cofactor peptides used: human SRC1 fragment (residues 686–698, RHKILHRLLQEGS) comprising the second nuclear receptor-interacting motif (Thermo Fisher), human NCOR1 fragment (residues 2554–2283, FADPASNLGLEDIIRKALMGSFDDKVEDHG) comprising the second nuclear receptor-interacting motif (LifeTein, LLC). A fully deuterated sample was generated by incubating protein with 4 M guanidine-HCl in D₂O for 30 min at 45 °C. Exchange reactions were performed with a CTC PAL sample handling robot (LEAP Technologies). Labeling times were 30, 60, 180, 600, and 1800 s in the kinetics experiments. Differential HDX of a complete compound selection was measured after 30 and 600 s of labeling time points. Reactions were conducted by incubating 4 µl of LXR with 50 µl of D₂O buffer (containing 25 mM Tris-HCl (pD 7.6), 50 mM NaCl, 10% glycerol, and 1 mM TCEP). The exchange reaction was stopped by the addition of 50 µl of quench solution (3 M urea, 0.1% TFA) followed by immediate injection into an Enzymate BEH Pepsin 2.1 × 30 mm Column (Waters, UK) for protein digestion (2 min at 24 °C). Peptic digest was further injected on a Waters nanoACQUITY UPLC System where peptides were first desalted by trapping for 3 min on a VanGuard Pre-Column Acquity UPLC BEH C18 (1.7 µm; 2.1 × 5 mm), and then eluted over a 6-min 5–40% (vol/vol) acetonitrile (containing 0.1% formic acid) gradient into a Waters Synapt G2-Si mass spectrometer. Peptide separation was conducted at 0.1 °C. All exchange reactions were performed in triplicates.

Peptides from three samples of non-deuterated LXR were identified by using the ProteinLynx Global Server (PLGS Waters, UK). Peptides with an intensity of over 5000, a mass error < 5 ppm, and present in at least two of the three data acquisitions were pooled and imported into the DynamX data analysis software (Waters, UK). After a first round of automated spectral processing by using DynamX, each peptide was inspected manually for suitability for further analysis.

For pairwise comparison, MEMHDX software was used to identify statistically significant changes of deuterium uptake[68]. For all peptides, a value of relative exchange per amino acid was corrected to back exchange. The mean deuteration level per amino acid was calculated by using Matlab (Mathworks)[69] and subsequently mapped onto the crystal structures with PyMOL (Schrodinger LLC). For structural representation, loop modeling with MOE (Chemical Computing Group Inc.) was applied on 2ACL.pdb file for LXRα LBD, and 1PQC.pdb for LXRβ LBD.

Data used for the multivariate analysis were collected in several rounds, apo protein was measured for each series of experiments, and used as a reference standard.

**Cofactor recruitment assay by using surface plasmon resonance.** Cofactor recruitment by LXRα and LXRβ was determined by using surface plasmon resonance (SPR) biosensor (Biacore T200, GE Healthcare) by probing the level of receptor binding to the surface-immobilized cofactor peptides (SRC1 (677–700): GSGSGSPSSHSSLTERHKILHRLLQEGSPS, NCOR1 (2251–2273): GSGSGSGHSFADPASNLGLEDIIRKALMG). The peptides carried an N-terminal biotin for surface immobilization on a Xantec NID200M Sensor Chip (NTA-derivatized carboxymethyldextran hydrogel, XanTec bioanalytics GmbH) pre-loaded with Ni²⁺. Petides were immobilized via His-tagged streptavidin (Abcam), to levels of 1000 ± 300 RU. LXRα and LXRβ LBD proteins were serially diluted in SPR running buffer (10 mM HEPES, 150 mM NaCl, and 0.05% Tween 20, pH 7.4) to a broad range of concentrations (0.02–10 µM) and equilibrated with tested compound (50 µM) prior to the sample injection over a modified biosensor surface. LXR-ligand complexes were injected in quick succession for 120 s over immobilized cofactor peptides and reference surfaces along with blank samples followed by 900 s of dissociation. The binding level at equilibrium was monitored for all samples, including the reference sample that contained DMSO at equal amount as tested compounds. All data were normalized to this reference sample in order to quantify the effects on peptide binding without bias by receptor instability. The surface was regenerated with 0.5% (w/v) SDS after each injection. Binding responses in the steady-state region of the sensorgrams were plotted against LXR LBD concentration to determine the overall equilibrium-binding affinity by fitting the data to nonlinear regression (Prism 8; GraphPad).

**Statistics and reproducibility.** Two principal multivariate statistical tools were used to analyze multidimensional HDX-MS data: principal component analysis (PCA), and orthogonal projections to latent structures (OPLS). SIMCA 15 (Sartorius Stedim Data Analytics AB) was used for multivariate analysis. The unsupervised PCA score plot was used for an overall visualization of measured compounds based on their HDX profiles and to observe whether compounds demonstrating similar in vivo properties were forming statistically distinct clusters. OPLS analysis was used to build the regression model between the multivariate HDX data and a response variable, either continuous ($pEC_{50}$) or binary containing class information—in the latter case the discriminant analysis (OPLS-DA) was

used. HDX data from 17 compounds were used to build X-matrix. For every compound, the difference in relative fractional uptake between individual peptides from ligand-bound and apo forms of both LXRα and LXRβ was measured, at two time points (30 and 600 s of deuterium exchange). This resulted in a HDX matrix comprising $(115 + 103) \times 2 = 436$ variables. Prior to the analysis, the HDX matrix was trimmed of missing data and data were mean-centered. The resulting models were tested for a goodness of fit ($R^2Y$) and predictive ability ($Q^2$) by using a sevenfold cross-validation strategy, CV-ANOVA test, and a permutation analysis of 100 iterations built in SIMCA 15. S plot was used for the selection of the most discriminant peptides between compound classes according to their correlation ($|p(corr)| > 0.7$) and contribution ($p > 0.05$) coefficients. A two-tailed Student's $t$ test was performed to verify that the peptides identified by OPLS-DA models were significantly different between the two classes of ligands (Prism 8; GraphPad).

**Reporting summary**. Further information on research design is available in the Nature Research Reporting Summary linked to this article.

## Data availability

Atomic coordinates and the related structure factors have been deposited in the Protein Data Bank with accession codes: 6S4T (LXRβ in complex with WAY-254011), 6S4U (LXRβ in complex with AZ3), 6S4N (LXRβ in complex with AZ6), and 6S5K (LXRβ in complex with AZ8). All other data generated or analyzed during the study in this published article (and its supplementary information files) are available upon reasonable request.

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

## Acknowledgements

The authors would like to thank AstraZeneca colleagues who have contributed to the generation of functional data used in this study that enabled the multivariate analysis, and specifically Rebecca Rae, Tineke Papavione, and Petra Johannesson for support and helpful discussions. Further, we want to thank Margareta Ek, Anna Jansson, and Jens Petersen for X-ray crystallography work. We are also grateful to Ewa Nilsson and Niek Dekker for protein supply, and to Anders Gunnarsson for assistance in SPR. A.Y.B. is a fellow of the AstraZeneca R&D postdoc program. I.G.S. acknowledges support from the American Heart Association (15GRNT25560038) and the National Institutes of Health (1R56HL131779-01).

## Author contributions

A.Y.B. carried out HDX-MS and SPR experiments and multivariate analysis. J.S. analyzed and prepared for publication of the X-ray structures of LXRβ complexes. E.B. was a part of the team who synthesized compounds used in this study. I.M. and I.G.S. carried out animal studies. D.H. carried out the modeling of in vivo data. A.Y.B., E.E., P.A. and E.L.L. designed and supervised the study. A.Y.B. wrote the paper. E.E., D.H., J.S., I.G.S., P.A. and E.L.L. contributed to paper preparation. All authors have given approval to the final version of the paper.

## Competing interests

The authors declare no competing non-financial interests but the following competing financial interests: all authors, except for I.G.S., are employees (and stockholders) of AstraZeneca UK Ltd. or were at the time that this study was conducted.
