## [Peer Review File · Communications Biology]

Reviewers' comments:

Reviewer #1 (Remarks to the Author):

Ligands targeting the LXR nuclear receptors show beneficial effects in the treatment of cardiovascular disease, but also display undesired side effects due to upregulated lipogenesis. In this manuscript, the authors tackle an outstanding question: on the molecular level, how do compounds with improved therapeutic profiles differentially affect the structural conformation of LXRs to elicit such effects? To address this, the authors perform differential HDX-MS structural analysis on LXRalpha and LXRbeta using a collection of LXR compounds and make structure-function correlations using sophisticated statistical methods. The data indicate that compounds that do not stabilize the the AF-2 surface (including helix 12) and a few other regions are associated with gene transcription signatures related to reduced side effect profiles.

Overall, this is an interesting area of research. Studies of other nuclear receptors, including PPARgamma, have shown similar profiles where compounds with more favorable therapeutic profiles do not stabilize the AF-2 surface. However, this study, which focuses on differential structure-function analyses of LXRalpha and LXRbeta, is novel and uses a robust and unbiased statistical framework that to my knowledge has not been employed in the study of other nuclear receptors.

I support publication of this manuscript in in Communication Biology and have relatively minor comments:

1. Some of the Supplementary Figures are called out of order
2. In Supplementary Figures 8 and 9, it would be helpful if the important secondary structural elements (e.g. helix 12, etc.) were noted on at least one structure.
3. On page 8, it is stated that LXRbeta is better stabilized by most of the compounds compared to LXRalpha. I have a few comments on this.

(3a) Is the apo-LXRbeta ligand-binding domain more dynamic than apo-LXRalpha?

(3b) A bacterial lipid from the host expression system (E. coli) were bound to the proteins to different degrees could influence the HDX-MS results. The methods section does not include any protein delipidation. Have the authors tested whether delipidation of the protein influences the HDX-MS results, which could in principle affect the conclusions of the manuscript on the LXRalpha vs. LXRbeta selectivity of the results?
4. On page 10, the authors state that "Unliganded LXR α and LXR β did not interact with the coactivator peptide..." but should add "under the conditions tested". The highest protein concentration used for the experiments was 10 μ M, and likely binding would be observed if the protein concentration was taken higher than 10 μ M.
5. On page 10, the authors refer to the non-lipogenic compounds as partial agonists based on the SPR data. Is there any published luciferase reporter transcription or gene expression data to support the classification of these compounds as partial (vs. full) agonists that could be cited?
6. On page 16, last sentence: change "regons form" to "region forms"
7. On page 17, the non-helix 12 mechanism of action for PPARgamma partial agonists is not only

associated changes in coregulator recruitment but also in changing the posttranslational modification status of PPARgamma (see PubMed IDs 20651683 and 21892191).

8. It would be useful to the field if the raw data used for the statistical analyses (HDX and functional data) were made available, as well as any scripts from the SIMCA 15 multivariate analysis, as this would help others in the field to learn how to use the robust statistical approach employed in this manuscript.

Reviewer #2 (Remarks to the Author):

the paper is well written and designed if we only take account into the LXR LBD domain, I guess the expression and purification of full length LXR is the problem although the full length structure has been solved.

In this study, compounds were well characterized for their cellular and functional activities and were subjected to HDX analysis. It is very similar to what has been done for other receptors like VDR, PPARg, and RORG. One shortcoming of this study is only two points of HDX data were collected and presented. In this study it seems the protein-ligand interactions are very dynamic across different time spans thus more time points were welcomed and recommended if they really want to differentiate the HDX kinetics between compounds. The receptor dynamics do not look like they require very long HDX incubation.

The Shared and Unique Structures (SUS) plot analysis is a very good way to present the local HDX dynamics data and how it correlates to the functions of various compounds. I am positive towards publication of this work.

Reviewer #3 (Remarks to the Author):

In their manuscript, Belorusova and colleagues report on a study aimed at identifying specific regions of the receptors LXR α / β whose differential stabilization would allow the separation of beneficial anti-atherogenic from adverse lipogenic effects and thus help the design of drugs with improved therapeutic indexes. The authors use a combination of experimental approaches, including hydrogen/deuterium exchange coupled with mass spectrometry, in vitro (ligand binding and transactivation assays, coregulator recruitment assays by SPR) and in vivo (ligand-induced elevation of triglyceride as the negative marker and induction of the cholesterol transporter ABCA1 gene as a positive marker) functional analyses or X-ray crystallography. These data are then analyzed by various statistical approaches in order to find significant correlations between the specific LXR regions stabilized by a given ligand and its pharmacological profile in terms of anti-atherogenic and lipogenic activities. Overall, the study addresses a very timely, interesting and challenging problem: how to separate the beneficial activities from the adverse effects inherent to all nuclear receptor ligands used as drugs. The manuscript contains a significant amount of data, it is well written and illustrated and the experiments appear to be adequately performed.

My main concern is about the mechanistic interpretation of the data which might not be as straightforward as presented in the manuscript. Indeed there are some inconsistencies between the

various experimental approaches and their interpretations. For example, the main conclusion of the study is that the negative effect (increase of plasma TG levels) of LXR agonists is mainly driven by the stabilization of helix H12 in its active (agonistic) position, whereas the positive effect (induction of ABCA1) is correlated with a stabilization of helices H3/H5. However, the authors report on a previous study where they found that the LXR agonist AZ876 inhibits the progression of atherosclerosis in mice without inducing liver steatosis (page 4 line 101-103). This observation does not support the conclusions of the manuscript since Table 1 shows that this ligand is a strong agonist (EC50 in the nanomolar range), in line with HDX-MS data showing that this ligand strongly stabilizes H12. Hence, the link between stabilization of the active conformation and TG induction is not so clear. Other intriguing observations are that some ligands (e.g. AZ5 and F1) act as agonists in the reporter gene assays but do not stabilize H12.

The authors explain that formation of a hydrogen bond between the ligand and residue H435 is critical for the stabilization of the agonistic conformation of H12, however it appears that 24,25EC and BMS-852927 which do not exhibit agonistic activity in transactivation assays form a hydrogen bond (mediated by a water molecule in the latter case) with H435, suggesting that this feature is not the only one at play in the stabilization of H12.

Regrettably, no attempt is made to provide a structure-based explanation for the differential impact of ligands on the stabilization of the transcriptionally active conformation (beside the hydrogen bond with H435) and helix H3. Since one of the aims of the study is to provide guidelines for the development of better LXR-based drugs, an in-depth structural analysis correlating the chemical structures, their interactions with the LXR ligand-binding pockets and HDX-MS data would greatly enhance the impact of the manuscript.

In the case of BMS-852927 that acts as an antagonist, can the authors discriminate between the ligand-induced stabilization of H3, and the masking of this helix by the positioning of helix H12 in the coregulator groove, thereby reducing H/D exchange?

It is unclear how BMS-852927 can act as an inverse agonist (stabilize the LXR/corepressor complex) and induce ABCA1 expression. It is also very surprising that none of the ligands, and even strong agonists, is able to dissociate NCoR from LXR α (Fig. 5). Could the authors comment on that? In the same line, the discussion p19 on how non-agonists can still promote gene expression is not convincing.

Since the experiments measuring ABCA1 levels are made in vivo, and since BMS-852927 is an inverse agonist that do not promote an active conformation of the receptor, is there any evidence that the elevation of ABCA1 is truly (an only) LXR-mediated in mice?

An information that is missing in the manuscript is the efficacy of ligands in the transactivation assays. In addition to their potencies (EC50), the efficacy values are important to characterize the various ligands as full or partial agonists.

We thank the Reviewers for dedicating their time into reading and evaluating our manuscript. We appreciate their constructive comments which we address in this point-by-point response. In our revised manuscript we modified the Table 1 and added the efficacy of ligands in transactivation assays as requested by Reviewer 3. We also changed the Supplementary Figure 6 to include three additional compounds for which the full HDX-MS kinetics has been measured, and the Supplementary Figure 11 to indicate hydrogen bond distances and interactions between the LXR β C-terminus and GW3965 or BMS-852927. Order of the Supplementary Figures 7-9 was corrected in accordance with their callouts in the manuscript.

Reviewers' comments:

Reviewer #1 (Remarks to the Author):

Ligands targeting the LXR nuclear receptors show beneficial effects in the treatment of cardiovascular disease, but also display undesired side effects due to upregulated lipogenesis. In this manuscript, the authors tackle an outstanding question: on the molecular level, how do compounds with improved therapeutic profiles differentially affect the structural conformation of LXRs to elicit such effects? To address this, the authors perform differential HDX-MS structural analysis on LXRalpha and LXRbeta using a collection of LXR compounds and make structure-function correlations using sophisticated statistical methods. The data indicate that compounds that do not stabilize the AF-2 surface (including helix 12) and a few other regions are associated with gene transcription signatures related to reduced side effect profiles.

Overall, this is an interesting area of research. Studies of other nuclear receptors, including PPARgamma, have shown similar profiles where compounds with more favorable therapeutic profiles do not stabilize the AF-2 surface. However, this study, which focuses on differential structure-function analyses of LXRalpha and LXRbeta, is novel and uses a robust and unbiased statistical framework that to my knowledge has not been employed in the study of other nuclear receptors.

I support publication of this manuscript in in Communication Biology and have relatively minor comments.

We thank the reviewer for their evaluation of our work and for comments listed below, which we have addressed in the revised manuscript.

1. Some of the Supplementary Figures are called out of order

We corrected the order of the Supplementary Figures.

2. In Supplementary Figures 8 and 9, it would be helpful if the important secondary structural elements (e.g. helix 12, etc.) were noted on at least one structure.

Secondary structural elements are now added to facilitate the figure interpretation.

3. On page 8, it is stated that LXRbeta is better stabilized by most of the compounds compared to LXRalpha. I have a few comments on this.

(3a) Is the apo-LXRbeta ligand-binding domain more dynamic than apo-LXRalpha?

The apo forms of LXR α and LXR β LBDs demonstrate relatively similar conformational dynamics (see *Figure below*) except for some regions. The most striking difference was observed for the C-terminus of H1 and the region connecting H1 and H3, which appear to be highly dynamic in LXR α while being relatively rigid in LXR β . Other regions more protected in LXR β comprise H3 which we discuss in the manuscript, as well as the loop between helices H8 and H9. LXR α , in turn, is less dynamic than LXR β in the loop between helices H7 and H8. Of note, among these regions only H3 is directly involved in formation of the ligand binding pocket and is differentially perturbed upon ligand binding. Despite the similarity between the ligand-binding pockets of LXR α and LXR β , we see a stronger reduction in deuterium exchange of LXR β upon compound binding (Figure 2 and Supplementary Table 1).

(3b) A bacterial lipid from the host expression system (*E. coli*) were bound to the proteins to different degrees could influence the HDX-MS results. The methods section does not include any protein delipidation. Have the authors tested whether delipidation of the protein influences the HDX-MS results, which could in principle affect the conclusions of the manuscript on the LXR α vs. LXR β selectivity of the results?

We have measured the HDX of LXR α and LXR β LBDs subjected to delipidation (1 h incubation with an equal volume of Hydroxyalkoxypropyl-Dextran (Sigma-Aldrich, Inc.) at room temperature), however did not observe any significant difference in solvent exchange between delipidated and non-delipidated proteins (*see Figure below*). As indicated in the previous comment, ligand-binding pockets of LXR α and LXR β in apo states as well as their AF-2 display similar conformational dynamics, indicating that even if purified proteins still contain tightly bound endogenous lipids and fatty acids not removed by chromatography, this is unlikely to contribute to the differences in HDX observed between the two isoforms upon binding of synthetic ligands. PPAR γ LBD has been co-crystallized with bacterial fatty acid binding at the orthosteric site (Shang et al. 2018), however no co-crystal structures of LXRs with endogenous lipids have been reported.

4. On page 10, the authors state that "Unliganded LXR α and LXR β did not interact with the coactivator peptide..." but should add "under the conditions tested". The highest protein concentration used for the experiments was 10 μ M, and likely binding would be observed if the protein concentration was taken higher than 10 μ M.

The text has been modified to specify that we did not observe apo LXR α and apo LXR β interactions with the SRC1 peptide under the conditions tested.

5. On page 10, the authors refer to the non-lipogenic compounds as partial agonists based on the SPR data. Is there any published luciferase reporter transcription or gene expression data to support the classification of these compounds as partial (vs. full) agonists that could be cited?

We have now included the agonism efficacy of tested compounds in the reporter gene assay, as was asked by Reviewer 3. We do not see the correlation between the compound lipogenicity and the agonist efficacy in our Gal4 reporter gene assay in U2OS cells, however there is published data for some of the low lipogenic compounds supporting our classification. BMS852927 was characterized as a partial LXR β -selective agonist in CV-1 cells in the original report (Kick et al. 2016); LXR-623 was reported to be partial agonist for both isoforms in Huh cells (Wrobel et al. 2008). GW3965 is usually classified as a full pan-agonist of LXR (Collins et al. 2002), however there is some evidence that its activity largely depends on the cell cofactor environment, for example it functions as a weak partial agonist of LXR in HepG2 but as a full agonist in HEK293 cells (Miao et al. 2004). In general, LXR ligands are known to display different agonist efficacies depending on the assay, e.g. which cell lines or the receptor construct were used. Using native reporter systems to grade LXR agonism could give more accurate data in comparison to Gal4 systems due to correct RXR/LXR heterodimerization.

6. On page 16, last sentence: change “regions form” to “region forms”

The sentence has been changed.

7. On page 17, the non-helix 12 mechanism of action for PPARgamma partial agonists is not only associated changes in coregulator recruitment but also in changing the posttranslational modification status of PPARgamma (see PubMed IDs 20651683 and 21892191).

We thank the reviewer for pointing out these studies which show that non-H12 mechanism of action can have other important implications in the nuclear receptor signaling. We modified the sentence and cited the above references.

8. It would be useful to the field if the raw data used for the statistical analyses (HDX and functional data) were made available, as well as any scripts from the SIMCA 15 multivariate analysis, as this would help others in the field to learn how to use the robust statistical approach employed in this manuscript.

A standardized structure for raw data and an open data server for annotated HDX-MS data have not been agreed upon or established by the HDX-MS community (Masson et al. 2019), however we can make the raw files available to interested parties through personal file sharing. HDX-MS data used for the statistical analyses corresponds to the differences in relative fractional deuterium uptake between individual peptides from ligand-bound and apo forms of both LXR α and LXR β and is provided in full in the Supplementary Table 1 and the Supplementary Table 4. These tables can readily be used for the multivariate analyses as the X-variable matrix. For the functional data, either continuous or binary (class assignment) response variables can be used, both types are provided in the Table 1 and Table 2, respectively. Although Python scripts can be used in SIMCA 15 to automate/facilitate tasks, in this study we only used the graphical interface to perform the analyses and visualization. We have included a detailed description of data treatment parameters as well as model validation in the Methods section so that the analyses can easily be adapted by anybody interested in integration of the HDX-MS data and various types of functional assays.

Reviewer #2 (Remarks to the Author):

the paper is well written and designed if we only take account into the LXR LBD domain, I guess the expression and purification of full length LXR is the problem although the full length structure has been solved.

We thank the reviewer for their evaluation of our work. We have succeeded with the expression and purification of the full-length LXR heterodimers, however in the current study we focused on the LBDs to establish a robust and cost-effective approach integrating HDX-MS and functional screenings which can be applied to a larger selection of compounds as well as to other nuclear receptors.

In this study, compounds were well characterized for their cellular and functional activities and were subjected to HDX analysis. It is very similar to what has been done for other receptors like VDR, PPAR γ , and ROR γ . One shortcoming of this study is only two points of HDX data were collected and presented. In this study it seems the protein-ligand interactions are very dynamic across different time spans thus more time points were welcomed and recommended if they really want to differentiate the HDX kinetics between compounds. The receptor dynamics do not look like they require very long HDX incubation.

We thank the reviewer for this comment. Indeed, we observe differential conformational dynamics of both LXR α and LXR β in a region- and ligand-specific manner across different time spans. However, previous studies on nuclear receptors (Chalmers et al. 2007; Strutzenberg et al. 2019) and other proteins (Sheff et al. 2017) have shown that 1-2 exchange time points is sufficient for screening and analysis of protein-ligand interactions using HDX-MS. To minimize the data collection and analysis time, as well as the amount of required protein, we decided to choose several sensitive time points for our HDX-MS screening by evaluating the HDX kinetics of LXR α and LXR β LBDs in apo forms and bound to four different ligands (T0901317, AZ2, AZ7 and AZ876). In the initial Supplementary Figure 6 only the kinetics with T0901317 was shown, however we have now included three other compounds for better visualization of our rationale. We have performed a Principal Component Analysis of the HDX-MS data collected for this initial set of compounds after 30, 60, 180, 600, 1800 and 3600 sec of deuterium exchange to identify time points which show maximal differences between the compounds (*Scores plots in the Figure below*) but covering the largest variety of peptic peptides contributing to these differences the most (*Loadings plots in the Figure below*). We observed similar differences between the compounds at early time points (30, 60 and 180 sec), and selected 30 sec time point based on the dynamic behavior of the C-terminus containing functionally important H12. Similarly, longer exchange times (600, 1800 and 3600 sec) also displayed similar patterns of peptide contribution, in particular for LXR α . 600 sec time point was chosen to capture the long-term stabilization of LXRs by compounds as after 1800 and 3600 sec contribution of LXR β peptides was minimal.

The Shared and Unique Structures (SUS) plot analysis is a very good way to present the local HDX dynamics data and how it correlates to the functions of various compounds. I am positive towards publication of this work.

We appreciate the reviewer's comment about the utility of the SUS plot and we hope that this will be of interest and use to the wide scientific community.

Reviewer #3 (Remarks to the Author):

In their manuscript, Belorusova and colleagues report on a study aimed at identifying specific regions of the receptors LXR α / β whose differential stabilization would allow the separation of beneficial anti-atherogenic from adverse lipogenic effects and thus help the design of drugs with improved therapeutic indexes. The authors use a combination of experimental approaches, including hydrogen/deuterium exchange coupled with mass spectrometry, in vitro (ligand binding and transactivation assays, coregulator recruitment assays by SPR) and in vivo (ligand-induced elevation of triglyceride as the negative marker and induction of the cholesterol transporter ABCA1 gene as a positive marker) functional analyses or X-ray crystallography. These data are then analyzed by various statistical approaches in order to find significant correlations between the specific LXR regions stabilized by a given ligand and its pharmacological profile in terms of anti-atherogenic and lipogenic activities. Overall, the study addresses a very timely, interesting and challenging problem: how to separate the beneficial activities from the adverse effects inherent to all nuclear receptor ligands used as drugs. The manuscript contains a significant amount of data, it is well written and illustrated and the experiments appear to be adequately performed.

We thank the reviewer for their evaluation of our work and for comments listed below, which we have tried to address and better clarify.

My main concern is about the mechanistic interpretation of the data which might not be as straightforward as presented in the manuscript. Indeed there are some inconsistencies between the various experimental approaches and their interpretations. For example, the main conclusion of the study is that the negative effect (increase of plasma TG levels) of LXR agonists is mainly driven by the stabilization of helix H12 in its active (agonistic) position, whereas the positive effect (induction of ABCA1) is correlated with a stabilization of helices H3/H5. However, the authors report on a previous study where they found that the LXR agonist AZ876 inhibits the progression of atherosclerosis in mice without inducing liver steatosis (page 4 line 101-103). This observation does not support the conclusions of the manuscript since Table 1 shows that this ligand is a strong agonist (EC50 in the nanomolar range), in line with HDX-MS data showing that this ligand strongly stabilizes H12. Hence, the link between stabilization of the active conformation and TG induction is not so clear. Other intriguing observations are that some ligands (e.g. AZ5 and F1) act as agonists in the reporter gene assays but do not stabilize H12.

A substantial interest in development of potent LXR agonists such as AZ876 is due to the idea that these ligands may still reduce atherosclerosis when given at low doses that do not provoke adverse effects. This strategy to improve the therapeutic window of LXR ligands was developed in parallel with discoveries of weak LXR agonists that increase *Abca1* mRNA above basal level and thus reduce atherosclerosis but display reduced side effects. In the original study it was demonstrated that AZ876 is a very potent LXR agonist which inhibits the progression of atherosclerosis in mice without inducing liver steatosis when given in low dose of 5 $\mu\text{mol}\cdot\text{kg}^{-1}\cdot\text{day}^{-1}$ (van der Hoorn et al. 2011). Nevertheless, high-dose AZ876 (20 $\mu\text{mol}\cdot\text{kg}^{-1}\cdot\text{day}^{-1}$) increased triglyceride levels by 110% in comparison to GW3965 which increased plasma triglycerides by 70% at a similar dose. In the current study we wanted to compare the intrinsic lipogenic properties of compounds independent of their pharmacokinetic properties and therefore assessed induced plasma TG levels as a function of plasma compound concentration rather

than at a particular dose. Since AZ876 increases TGs when present in plasma at much lower concentrations than other compounds, we classified AZ876 as lipogenic .

We agree that the observation that LXR α H12 is not stabilized by agonist ligands when measured by HDX-MS is intriguing. Given the fact that H12 is highly dynamic, it is possible that the stabilization effect was not observed under conditions tested, and perhaps shorter labelling times could be used in the HDX-MS experiments. Agonist compounds, however, are still able to induce coactivator peptide recruitment and to stabilize the LXR α AF-2 measured by HDX-MS in the presence of the SRC1. The observation that use of the LXR/SRC1 HDX-MS dataset leads to a significant improvement of the TG multivariate model strengthens our findings that negative lipogenic effects of LXR ligands are largely H12- and coactivator-driven.

The authors explain that formation of a hydrogen bond between the ligand and residue H435 is critical for the stabilization of the agonistic conformation of H12, however it appears that 24,25EC and BMS-852927 which do not exhibit agonistic activity in transactivation assays form a hydrogen bond (mediated by a water molecule in the latter case) with H435, suggesting that this feature is not the only one at play in the stabilization of H12.

The Reviewer correctly mentioned that 24,25EC and BMS-852927 do not exhibit any agonistic activity in our transactivation assay, however we refer the readers to earlier studies where they were described as partial agonists (Williams et al. 2003; Kick et al. 2016). This discrepancy could originate from differences in cell types used, amount of transfected plasmids etc. Furthermore, in the reported study BMS-852927 agonist activity was determined using the full-length LXR constructs contrary to our assay where Gal4 DBD-LXR LBD chimeras were used. As the LXR-RXR heterodimer has been shown to be transcriptionally activated by heterodimerization itself (Wiebel & Gustafsson 1997), the presence of endogenous RXR could result in observed agonistic properties of BMS-852927.

Among ligands interacting with the LXR β His435-Trp457 activation switch via direct hydrogen bonding as observed in crystal structures, 24,25EC displays the weakest protection of the LXR β H11-H12 from solvent exchange when measured by HDX-MS: it reduces exchange by 10% after 30 sec of labelling, which is significantly less efficient than other compounds directly interacting with His435 (AZ3, AZ6, AZ8, T0901317, WAY-254011) that reduce HDX of H11-H12 by 17-25%. Analysis of crystal structures reveals that His435 is in a suboptimal position in the 24,25EC complex at a distance of 3.4 Å from the epoxide of the sterol side chain and thus forms a weak hydrogen bond with the ligand (Williams et al. 2003). Other agonists form shorter hydrogen bonds of 2.5-3.1 Å essential for proper position of H12, which correlates with their activities both in HDX-MS and functional assay. We have added the hydrogen bond distances to the Supplementary Figure 11 and modified the manuscript text accordingly.

When looking at the ligand-induced stabilization of LXR β H12 observed in HDX-MS, BMS-852927 and GW3965 display the weakest perturbation effects on H12. As we mention in the manuscript, these are the only ligands not forming a (direct) hydrogen bond with H435; however, in the case of GW3965 its chloro-trifluoromethylbenzyl group is involved in multiple hydrophobic interactions with Val439, Leu442, Leu449, Leu453 and Trp457 which partially compensates for the hydrogen bonding loss (*interactions between ligands and residues of LXR β H11-H12 at 4.5 Å cutoff are shown in the Figure below*). For the BMS-852927 only two out of four LXR β molecules present in the asymmetric unit contain H12 in its active conformation, and in those complexes H11 is shifted away from the ligand by 3.5 Å (*right panel of the Figure below*) which leads to the loss or weakening of the interactions with His435,

Val439, Leu442 and Leu 449. We thank the reviewer for the comment on H12 stabilization, we have modified the manuscript text to clarify how other interactions contribute to the proper positioning of H12 and added panels to the Supplementary Figure 11.

Regrettably, no attempt is made to provide a structure-based explanation for the differential impact of ligands on the stabilization of the transcriptionally active conformation (beside the hydrogen bond with H435) and helix H3. Since one of the aims of the study is to provide guidelines for the development of better LXR-based drugs, an in-depth structural analysis correlating the chemical structures, their interactions with the LXR ligand-binding pockets and HDX-MS data would greatly enhance the impact of the manuscript.

We absolutely agree with the Reviewer that such an in-depth structural analysis would increase the impact of our work. Unfortunately, we did not uncover the atomic details of differential stabilization of the LXR α helix H3 by the two classes of ligands (high inducers of *Abca1* vs. low inducers) from the available LXR β structures. Similarly, others have not been able to identify structural reasons for the LXR β over LXR α selectivity of LXR ligands from crystal structures even when structures with both receptor isoforms were obtained (Kick et al. 2016; Stachel et al. 2016). NMR analysis could help reveal atomic details of differential ligand interactions, but such study is out of the scope of the present paper and could be considered in our future work. Nevertheless, we believe that our manuscript contains adequate amount of data required to demonstrate that HDX-MS can be used to identify directions for the development of ligands with improved therapeutic profiles via identification of a list of the most discriminating peptides associated with positive pharmacological features of ligands, which can be further applied for predictive classification of ligands in subsequent higher throughput HDX-MS assays.

In the case of BMS-852927 that acts as an antagonist, can the authors discriminate between the ligand-induced stabilization of H3, and the masking of this helix by the positioning of helix H12 in the coregulator groove, thereby reducing H/D exchange?

Our HDX-MS data of both LXR α and LXR β LBDs in apo form indicates that H12 is highly dynamic/disordered in solution (Supplementary Figure 6f). No perturbation in H12 dynamics is observed upon binding of BMS-852927 and thus it is highly unlikely that H12 gets positioned in the coregulator groove when this ligand is bound.

It is unclear how BMS-852927 can act as an inverse agonist (stabilize the LXR/corepressor complex) and induce ABCA1 expression. It is also very surprising that none of the ligands, and even strong agonists, is able to dissociate NCoR from LXR α (Fig. 5). Could the authors comment on that? In the same line, the discussion p19 on how non-agonists can still promote gene expression is not convincing.

Agonistic/antagonistic activity of BMS-852927 is dependent on the particular assay. We only see it acting as an inverse agonist in the corepressor peptide recruitment assay which utilizes relatively high concentrations of receptor LBD and interacting peptides. In our preliminary results from the human whole blood assay we see that it induces ABCA1 mRNA levels, however with lower efficiency than full agonists.

From our cofactor peptide recruitment assay it seems that LXR α is more flexible and maintains a more dynamic equilibrium between agonistic and antagonistic conformations compared to LXR β , even when bound to a pure agonist. Interacting cofactor peptides help to stabilize either of the two conformations in an "induced fit"-like mechanism that may not reflect the physiological balance among ligands, coactivators and corepressors that is achieved in cells. Similar differences between corepressor peptides binding to the alpha and beta isotypes were observed in earlier studies, where partial agonists were unable to dissociate corepressor peptides from LXR α but could dissociate peptides from LXR β (Albers et al. 2006). We also want to stress that the peptides used in our study might not fully represent the context of the full-length corepressors, where other protein regions are likely to allosterically modulate the complex formation. Abca1 expression is well known to be regulated by corepressors NCOR and SMRT, and we hope that our work will promote further studies of the LXR/corepressor complexes. The role of LXR heterodimeric partner RXR in this process is also still unclear.

Since the experiments measuring ABCA1 levels are made in vivo, and since BMS-852927 is an inverse agonist that do not promote an active conformation of the receptor, is there any evidence that the elevation of ABCA1 is truly (an only) LXR-mediated in mice?

Numerous studies have shown that the murine and human *Abca1* genes are regulated by the LXR/RXR heterodimers (Costet 2000; Schwartz et al. 2000; Repa et al. 2000; Venkateswaran et al. 2000). PPAR γ was shown to cooperate with LXR α in the regulation of ABCA1 and ABCG1 expression, however the effects of PPAR γ ligands were likely to be secondary to induction of LXR α expression (Chawla et al. 2001). At any rate, we did not detect PPAR $\alpha/\gamma/\delta$ agonism when tested the compounds in the cell-based transactivation assay (AZ2-4 and AZ6-9 were profiled). Furthermore, BMS-852927 does not induce LXR target gene expression in LXR KO macrophages (see Figure below, please note that basal *Abca1* mRNA levels are elevated in vehicle treated KO macrophages because LXR-dependent repression is lost).

Mouse bone marrow derived macrophages treated with 1.0 μM BMS852927 for 18 hours.

An information that is missing in the manuscript is the efficacy of ligands in the transactivation assays. In addition to their potencies (EC50), the efficacy values are important to characterize the various ligands as full or partial agonists.

We thank the reviewer for pointing this out, the agonism efficacy of tested compounds in the reporter gene assay is included in the Table 1 in our revised manuscript.

REVIEWERS' COMMENTS:

Reviewer #1 (Remarks to the Author):

The authors adequately addressed all reviewer comments. I support publication.

Reviewer #3 (Remarks to the Author):

I am satisfied with the authors' response to my comments and I support publication of the manuscript in Communications Biology.